# Minimal exposure of lipid II cycle intermediates triggers cell wall antibiotic resistance

Hannah Piepenbreier[1], Angelika Diehl[1] & Georg Fritz [1]

Cell wall antibiotics are crucial for combatting the emerging wave of resistant bacteria. Yet, our understanding of antibiotic action is limited, as many strains devoid of all resistance determinants display far higher antibiotic tolerance in vivo than suggested by the antibiotic-target binding affinity in vitro. To resolve this conflict, here we develop a comprehensive theory for the bacterial cell wall biosynthetic pathway and study its perturbation by antibiotics. We find that the closed-loop architecture of the lipid II cycle of wall biosynthesis features a highly asymmetric distribution of pathway intermediates, and show that antibiotic tolerance scales inversely with the abundance of the targeted pathway intermediate. We formalize this principle of minimal target exposure as intrinsic resistance mechanism and predict how cooperative drug-target interactions can mitigate resistance. The theory accurately predicts the in vivo efficacy for various cell wall antibiotics in different Gram-positive bacteria and contributes to a systems-level understanding of antibiotic action.

[1] LOEWE Center for Synthetic Microbiology, Philipps-Universität Marburg, Hans-Meerwein-Strasse 6, 35032 Marburg, Germany. Correspondence and requests for materials should be addressed to G.F. (email: georg.fritz@synmikro.uni-marburg.de)

Theoretical modelling of key biological processes has advanced our understanding of how cells respond towards environmental perturbations, such as antibiotic treatment. For instance, in *Escherichia coli* mathematical modelling accurately predicted non-trivial susceptibility patterns against ribosome-targeting antibiotics at different growth rates[1], showed that a positive feedback on resistance gene regulation can lead to growth bistability of an *E. coli* population under chloramphenicol treatment[2], and revealed how non-optimal responses to DNA stress under ciprofloxacin treatment can lead to suppressive drug interactions when combined with ribosome-targeting antibiotics[3]. Jointly, these studies demonstrate that intricate interactions between well-characterised biological parts elicit emergent and sometimes counterintuitive physiological responses, which can hardly be understood without theoretical frameworks. However, to date most of the predictive models for drug-target interactions focussed on translation-inhibiting antibiotics, which is facilitated by a well-established theoretical framework describing ribosome partitioning within bacterial cells[4–6]. Thus, to gain a better understanding of antibiotics targeting other essential processes, such DNA synthesis, transcription, and cell envelope biogenesis, theoretical models for these essential processes are urgently needed.

Antibiotics targeting the cell wall biosynthetic pathway are amongst the most important, clinically relevant last-resort antibiotics, such as ramoplanin, vancomycin and other glycopeptides[7,8]. Despite decades of experimental studies of the cell wall biosynthetic pathway in various organisms, to date there remain significant gaps in our understanding of cell wall antibiotic action. Most strikingly, for many cell wall antibiotics there are vast differences between their in vivo efficacy compared to the in vitro binding affinity for their molecular target—even in strains deleted for resistance determinants that could reduce antibiotic potency in vivo. For instance, in mutants of *Bacillus subtilis*, *Staphylococcus aureus* and *Enterococcus faecalis* lacking all known resistance determinants against nisin, ramoplanin or vancomycin, the minimal inhibitory concentrations (MICs) against these antibiotics are 20–200-fold higher[9–14] than the dissociation constants ($K_D$) for the respective drug-target interaction[15–17], highlighting that these antibiotics are significantly less active in vivo than in vitro. This apparent in vivo efficacy gap led to the notion that either all of these organisms might carry additional, yet undiscovered resistance determinants, or that the antibiotics might be inactivated in vivo, e.g., via sequestration to auxiliary cellular structures, effectively reducing the concentration of active antibiotics[10,16,18]. The variety of compounds, as well as the diversity of species displaying an in vivo efficacy gap, raises doubts about these hypotheses and suggests that there might be another, more universal origin of this phenomenon.

A more parsimonious explanation for this gap could emerge from the complex dynamics of the cell wall biosynthetic pathway itself, which is highly conserved across the bacterial world (reviewed e.g. in [8,19,20]). At the core of this pathway is the lipid II cycle, which encompasses all membrane-associated reactions of cell wall biosynthesis and is responsible for shuttling peptidoglycan (PG) subunits across the cytoplasmic membrane (Fig. 1a). Briefly, MraY and MurG sequentially attach the PG precursors UDP-MurNAc-pentapeptide and UDP-GlcNAc to the lipid carrier undecaprenyl phosphate (UP), giving rise to the lipid I and lipid II intermediates, respectively. Various flippases translocate lipid II to the outer leaflet of the cytoplasmic membrane, where penicillin-binding proteins (PBPs) incorporate the subunits into the growing PG layer. The resulting pyrophosphorylated state of the lipid carrier (UPP) is dephosphorylated by UPP phosphatases (UppPs) to yield the initial substrate UP for another round of PG subunit transport. Given that these

cyclic reactions represent the rate-limiting step of cell wall biosynthesis, it is not surprising that a wide range of antibiotics act by blocking progression of the lipid II cycle. This is achieved by either targeting the activity of the involved enzymes, e.g. PBPs (inhibited by beta-lactams) and MraY (inhibited by tunicamycin), or by directly sequestering the intermediate substrates of the lipid II cycle, e.g. UP (sequestered by friulimicin), UPP (sequestered by bacitracin) or lipid II (sequestered by ramoplanin, vancomycin and nisin), see Fig. 1a for an illustration and[8,20] for reviews.

To gain a quantitative understanding on how cell wall antibiotics interfere with this essential pathway, we here set out to derive a detailed, computational model of the lipid II cycle. By incorporating experimentally determined parameters from the literature, our theory accounts for key biochemical knowledge of this pathway and reconciles it with the in vivo inhibition patterns under antibiotic treatment. In particular, by focussing on the Gram-positive model organism *Bacillus subtilis*, we provide clues on the inner working mechanisms of cell wall biosynthesis and predict the in vivo efficacy of different cell wall antibiotics from first principles. In particular, we focus on antibiotics targeting different intermediates of the lipid II cycle (substrate-sequestering antibiotics), i.e. bacitracin, friulimicin, ramoplanin, vancomycin and nisin (Fig. 1a), which are active against a broad range of Gram-positive bacteria. Our results reveal that the in vivo efficacy gap is an emergent property of the lipid II cycle, leading us to suggest a novel principle of minimal target exposure as an intrinsic resistance mechanism towards substrate-sequestering cell wall antibiotics. Strikingly, our theory predicts that this intrinsic resistance can be circumvented—at least partially—by drugs that cooperatively bind their targets, providing a quantitative explanation for the pivotal role of cooperative binding for the potency of vancomycin and other glycopeptide antibiotics[21–23]. Thus, the theory presented here not only provides insights into the response of a universally conserved metabolic pathway towards perturbations, but also guides the design of novel antimicrobial compounds to efficiently block this core process of cell wall biosynthesis.

## Results

**Rationale of this study**. The bacterial cell wall consists of an alternating polymer of N-acetylglucosamine (GlcNAc) and N-acetylmuramic acid (MurNAc), cross-linked by a MurNAc-attached pentapeptide (Fig. 1a)[24,25]. Even though Gram-negative and -positive bacteria greatly vary in cell wall thickness and some organisms show specific modifications in peptidoglycan composition (e.g. variations in the GlcNAc-MurNAc-pentapeptide known for *Staphylococci*) or cross-linking properties (e.g. in *Corynebacteria*)[26], the central lipid II cycle of cell wall biosynthesis is highly conserved throughout the bacterial world (Fig. 1a). Accordingly, it seems plausible that the basic working principles of the lipid II cycle are similar between Gram-negative and Gram-positive bacteria. Most biochemical work on the enzymes and intermediates of the lipid II cycle, however, was focussed on the Gram-negative model organism *E. coli*. Therefore, in the following we will first perform some general considerations on the kinetics of cell wall synthesis in *E. coli*, which will lead us to a first quantitative model for this essential process in Gram-negatives. Given that most antibiotics targeting the intermediates of the lipid II cycle are ineffective against Gram-negatives (due to the permeability barrier posed by the outer membrane), we will adapt the model to Gram-positive-specific cell wall synthesis in a second stage. This will allow us to make testable predictions for cell wall antibiotic action in *B. subtilis* and other Gram-positive bacteria.

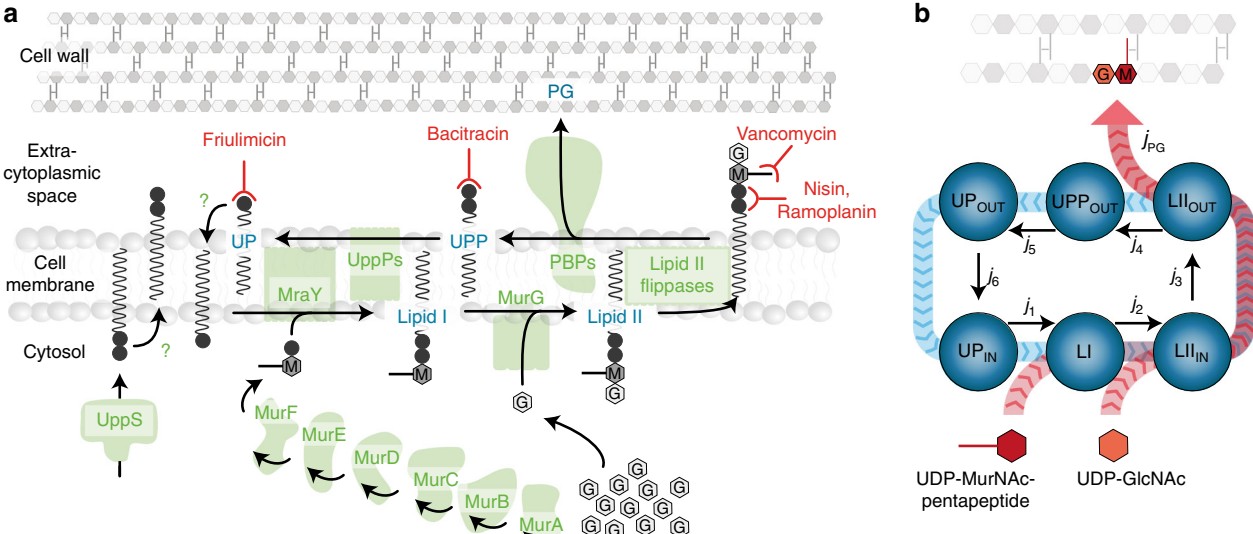

**Fig. 1** The lipid II cycle of Gram-positive bacteria is a prime target for antibiotics. **a** The lipid II cycle, as the core pathway of cell wall biosynthesis, drives the transport of PG subunits across the cytoplasmic membrane via attachment to lipid carrier molecules. The cytoplasmic production of UDP-MurNAc-pentapeptide (M) from UDP-GlcNAc (G) is catalysed by the MurA-F ligases[34,75,76]. Subsequently, at the internal leaflet of the cytoplasmic membrane the translocase MraY and the transferase MurG sequentially attach UDP-MurNAc-pentapeptide and UDP-GlcNAc to the lipid carrier undecaprenyl phosphate (UP), giving rise to the lipid I and lipid II intermediates, respectively. Various flippases translocate lipid II to the outer leaflet of the cytoplasmic membrane, where penicillin-binding proteins (PBPs) incorporate the subunits into the growing PG layer. This leaves the lipid carrier in its pyrophosphate form (UPP), which has to be recycled to UP by dephosphorylation to allow a new round of PG monomer transport. Given that all known UPP phosphatases (UppPs) act at the external leaflet of the cytoplasmic membrane[77,78], carrier recycling requires flipping of UP to the internal leaflet by a yet unknown mechanism[68,69]. Finally, dilution of lipid carriers is counterbalanced by cytoplasmic synthesis of UPP by UppS, but likewise to UP flipping, the required mechanism to present UPP to the externally acting phosphatases is unknown. Several antibiotics inhibit key steps of cell wall biosynthesis by forming complexes with UP, UPP or lipid II, as indicated by the T-shaped red lines. **b** The lipid II cycle can be considered as a closed-loop system, in which all fluxes $j_i$ from one state of the cycle into the next balance each other. Since UDP-MurNAc-pentapeptide and UDP-GlcNAc use lipid II cycle intermediates as carriers for the transport across the cytoplasmic membrane, the flux of PG precursors, $j_{PG}$ (red arrows), is equal to the flux of the cycling reactions (blue arrows)

**Physiological constraints on PG synthesis**. In a first step we wondered about the total demand of PG synthesis of a bacterial cell, and accordingly, how fast the lipid II cycle has to shuttle PG monomers across the cytoplasmic membrane. During bacterial growth the synthesis of the wall has to precisely match the volume expansion of the cell, and any misbalance induced by antibiotic inhibition can lead to destabilization and lysis of the cell[27,28]. Accordingly, given that the sacculus of E. coli contains $N = 3.5 \times 10^6$ PG monomers (at a doubling time $T_D = 36$ min)[29] and that $\sim\delta = 50\%$ of the produced PG is degraded by hydrolases[30,31], balanced growth requires that the total rate of PG monomer translocation across the membrane, $j_{PG}$, has to equal $j_{PG} = (1 + \delta)N\frac{\ln(2)}{T_D} \sim 10^5$ PG monomers per minute. This high rate of transmembrane transport is supported by attaching PG monomers to a limited number of $1.5 \times 10^5$ lipid carrier molecules[32,33]. At the required synthesis rate, this implies that each lipid carrier transitions within 90 seconds through all states of the cycle (UPP > UP > lipid I > lipid I> UPP >...) (see Supplementary Note 1 for detailed estimation). Thus, each carrier undergoes an average of ~24 transport cycles before it gets diluted due to cell growth. This suggests that instead of synthesizing lipid carriers for one-time "use-it and lose-it" transport, lipid carrier recycling is the pace-maker of PG monomer transport across the membrane. Under these conditions the lipid II cycle can be approximated as a closed-loop system, in which the pool levels of lipid II cycle intermediates quickly equilibrate, leading to cyclic flux-balance between all of the states, i.e. $j_1 = j_2 = ... = j_6$ (Fig. 1b, blue arrows; Supplementary Note 1 and Supplementary Fig. 1a, b). For instance, if one reaction is limited by either the catalytic rate or the abundance of the respective enzyme, the substrate of this reaction will accumulate and all other intermediates will deplete until all fluxes in the cycle are equal. Experiments in E. coli indeed revealed a highly asymmetric distribution of lipid II cycle intermediates, with a ~100-fold excess of UPP and UP ($1.2 \times 10^5$ and $0.3 \times 10^5$ molecules per cell, respectively)[33] over lipid I and lipid II (700 and 1000, respectively)[32] (Supplementary Table 1a). Also, it is noteworthy that under normal growth conditions cells homeostatically control cytoplasmic UDP-MurNAc-pentapeptide and UDP-GlcNAc at levels that saturate MraY and MurG, respectively[34,35] such that the rate of wall synthesis is not limited by soluble PG precursor abundance. Instead, under these conditions the total flux of PG subunits across the cytoplasmic membrane (Fig. 1b; red arrows) is only limited by the membrane-associated steps of wall synthesis and is identical to the individual, cyclic fluxes in the lipid II cycle, i.e. $j_{PG} = j_1 = j_2 = ... = j_6$.

**Kinetic model of the lipid II cycle**. Are the molecular properties of the cycle compatible with the overall demand for cell wall synthesis outlined above? To test this, we developed a detailed kinetic model of the lipid II cycle, which integrates key biochemical knowledge from literature and simulates the overall rate of PG synthesis, $j_{PG}$. Briefly, the model takes into account the reactions depicted in Fig. 1a, by considering Michaelis-Menten kinetics for all characterised enzymes, and first order kinetics in case of the flipping reactions for UP, UPP and lipid II, since less is known about the latter. By further assuming production of UPP in the cytoplasm and dilution of all cycle intermediates due to cell growth, the model describes the dynamic changes in the concentrations of cycle intermediates in the inner and outer leaflet of the membrane (see Supplementary Fig. 1c and Methods). To

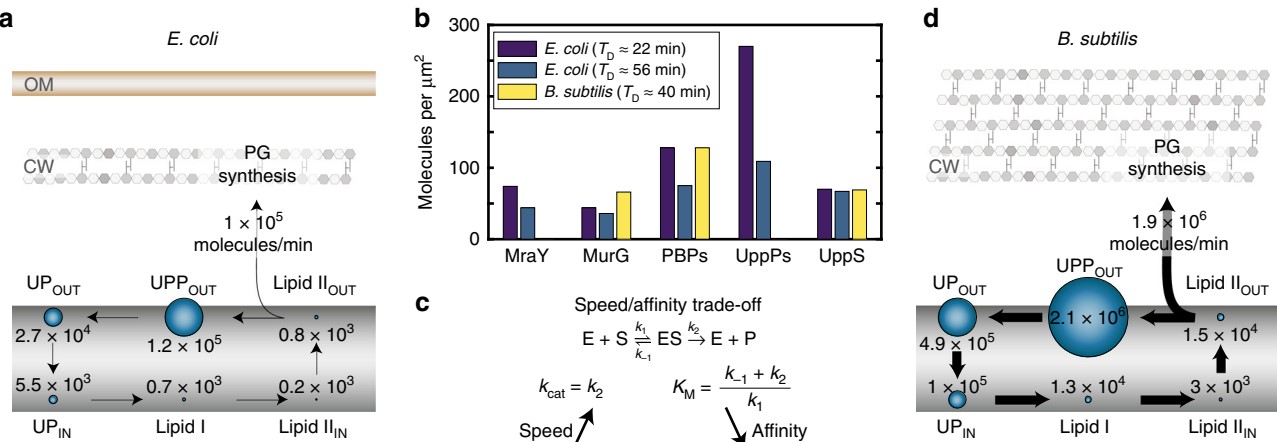

**Fig. 2** Abundance of enzymes and lipid carrier intermediates in the lipid II cycle. **a** Pool level distribution of lipid II cycle intermediates and rate of PG synthesis predicted by the theoretical model for *E. coli*. **b** The surface concentration (number of enzymes per unit surface area) of the PG synthesis machinery is similar in Gram-positive and -negative organisms. **c** Increased catalytic rates of the lipid II cycle enzymes are expected to significantly speed-up the PG synthesis in Gram-positive organisms. As the catalytic rate $k_{cat} = k_2$ also affects the Michaelis constant $K_M$, an increase in the speed of an enzymatic reaction can decrease the affinity of the enzyme for its substrate. **d** Pool level distribution of lipid II cycle intermediates and rate of PG synthesis predicted by the theoretical model for *B. subtilis*. The significantly thicker PG layer in *B. subtilis*, which compensates the lack of an outer membrane, demands an increase in the rate of PG synthesis, implying higher levels of lipid II intermediates shuttling faster through the cycle

calibrate the parameters in the model, we fixed all catalytic rates ($k_{cat}$) and Michaelis–Menten constants ($K_M$) to the values obtained from literature (Supplementary Table 1) and applied a constrained optimisation approach to estimate the remaining parameters (see Supplementary Note 1). In particular, by imposing that the overall flux within the lipid II cycle has to match the overall PG demand of the cell and by fixing the total abundances of cycle intermediates to the asymmetric distributions reported in literature (Supplementary Table 1a), we obtained precise estimates for the levels of the lipid II cycle-associated enzymes, as well as for the rates for lipid carrier flipping (Supplementary Fig. 1d). Interestingly, the theoretically predicted enzyme levels are in excellent agreement with a previous proteomics study[36] (Supplementary Table 2), showing that our model describes the quantitative dynamics of the lipid II cycle in a self-consistent manner—compatible with biochemical and physiological constraints.

When the flux across all reactions of the lipid II cycle is balanced, the model predicts an asymmetric distribution of cycle intermediates across the two leaflets of the cytoplasmic membrane. Especially, UPP and lipid II are predominantly found in the external leaflet, while UP displays an even distribution (Fig. 2a). Within the model, this is caused by highly efficient rates of UPP and lipid II flipping across the membrane, whereas the flipping of UP is predicted to be ~2 orders of magnitude slower. This is consistent with the fact that lipid II is actively transported from the internal to the external leaflet via MurJ and other flippases[37–39] and suggests that UPP could similarly follow an active transport route. In contrast, UP may follow a passive translocation process from the outer to the inner leaflet of the membrane (see Discussion). Taken together, this initial mathematical model for the lipid II cycle provides a first holistic view on this essential metabolic pathway in the Gram-negative model organism *E. coli*, integrating key biochemical properties, enzyme concentrations and pool levels of cycle intermediates.

Even though for Gram-positive bacteria a comprehensive biochemical understanding of the PG synthetic machinery, and in particular of the PBPs, has not been laid out, we next integrated all existing quantitative knowledge from diverse species to consolidate them in a modified mathematical model for the Gram-positive cell wall synthesis. First of all, while *E. coli* features

a PG thickness of 1.5 glycan layers on average[40], *B. subtilis* and many other Gram-positive bacteria have a much thicker wall of about 20 layers[8]. Thus, when comparing Gram-negative and Gram-positive cells of equal sizes and at similar doubling times, the lipid II cycle has to transport PG precursors at a ~13-fold higher rate in the latter (see Supplementary Note 1 and Supplementary Table 3a for a comparison between *E. coli* and *B. subtilis*). Theoretically, increases of the PG synthesis rate can be achieved by tuning three factors: (i) increasing the abundance of enzymes in the lipid II cycle, (ii) increasing the concentrations of lipid carriers, or (iii) increasing the catalytic rates of all associated enzymes. Interestingly, although proteomic studies in *B. subtilis* and *E. coli* revealed differences in the absolute enzyme abundances[36,41], their surface concentration is almost invariant between organisms—with typically between 50 and 100 molecules per µm² for each enzyme species (Fig. 2b and Supplementary Table 3b)—showing that Gram-positive bacteria do not simply increase the abundance of the PG synthetic machinery. Instead, in a range of Gram-positive bacteria the surface concentrations of the lipid carriers UP, UPP and lipid II are 10- to 20-fold higher compared to *E. coli* (Supplementary Table 3c), suggesting that these increased substrate levels are required to fully saturate the enzymes of the lipid II cycle in Gram-positives. Consistent with this, literature suggests that the $K_M$ value of MraY is eight-fold higher in *B. subtilis* ($K_M = 160$ µM[42]) compared to *E. coli* ($K_M = 20$ µM[43]). However, if the goal is to speed up PG synthesis—why does the Gram-positive PG synthetic machinery feature lower substrate affinity while increasing substrate abundance, ultimately leading to comparable levels of enzyme saturation as in Gram-negatives? A potential origin could lie in the speed/affinity trade-off known in enzyme kinetics[44,45], according to which speeding up the $k_{cat}$ value of an enzyme can lead to a sacrifice in substrate affinity and a concomitant increase of the $K_M$ value (Fig. 2c). For highly efficient enzymes, in particular, the $k_{cat}$ value is larger than the substrate dissociation rate $k_{-1}$, leading to an inverse relationship between affinity ($K_M^{-1}$) and speed, i.e., $K_M \approx k_{cat}/k_1$.

Taken together, the most parsimonious model for the Gram-positive lipid II cycle is that the ~13-fold higher demand for PG synthesis (compared to Gram-negative bacteria) is met by faster enzymes with 10–20-fold higher catalytic rates. The speed-affinity

trade-off then dictates that all substrate affinities will be 10–20-fold lower in Gram-positive enzymes, as observed for the $K_M$ value of MraY. This model is also consistent with the experimentally observed 10–20-fold higher lipid carrier substrate pools, which would then be required to achieve similar levels of enzyme saturation as in Gram-negatives, such that enzymes can operate close to their maximal speeds. Accordingly, to establish a self-consistent generic model for the Gram-positive cell wall biosynthesis, we scaled all parameters for the lipid II cycle in *E. coli*, i.e., the $k_{cat}$ and $K_M$ values, as well as the rate of UPP de novo biosynthesis, by a factor of 13 (see Supplementary Note 1). Accordingly, within this rescaled model both the overall PG synthesis rate, as well as all lipid carrier concentrations increase by this factor, while the relative stoichiometries between the lipid carrier intermediates remain identical to the model for *E. coli* (Fig. 2d). Although we are well aware that this coarse-grained scaling is an approximation for the lipid II cycle in *B. subtilis*, it is the most parsimonious choice of model parameters and leads to testable predictions for the cellular response towards cell wall antibiotics, as studied in the following.

**Predicting cell wall antibiotic action from first principles.** As introduced above, many cell wall antibiotics bind to externally exposed lipid II cycle intermediates, thereby sequestering lipid carriers from the cycle. For the five antibiotics under consideration (friulimicin, bacitracin, vancomycin, nisin and ramoplanin, see Fig. 1a) both the molecular targets as well as the equilibrium dissociation constants ($K_D$) for the antibiotic/target interaction have been characterised in vitro (Supplementary Table 4a). This allowed us to integrate these binding reactions into our quantitative model for the lipid II cycle (see Methods)—thereby creating a tool to generate predictions of cell wall antibiotic action without invoking further fit parameters. In the following we will first focus on the two antibiotics that bind their target non-cooperatively and later consider the effect of cooperative binding for the remaining three antibiotics.

First we studied the action of the cationic antimicrobial peptide bacitracin, which is widely used as a medicine and feed additive. Bacitracin binds to UPP by forming an amphipathic shell around its pyrophosphate group, thereby sequestering the target[46]. When incorporating the binding of bacitracin to UPP into our model ($K_D^{BAC} = 1\,\mu M$[47]) we predict a hyperbolic decrease of the total PG synthesis rate with increasing antibiotic concentration, reaching 50% of the maximal PG synthesis rate at 1.8 μM bacitracin ($IC_{50}^{BAC}$) (Fig. 3a). To understand why the predicted $IC_{50}$ almost coincides with the $K_D$ value in the model, we analysed the relative abundances of lipid II cycle intermediates at different bacitracin concentrations (Fig. 3b). Here it turned out that the $IC_{50}$ coincides with a decrease of the free external lipid II pool to approximately 50% of its untreated level, consistent with the role of lipid II as substrate for the final step of PG precursor incorporation. The reduction of free lipid II pools is correlated with an increase of the bacitracin-bound form of external UPP (commencing at the $K_D$ value), which effectively sequesters lipid carriers from the cycle and thereby triggers a concerted decrease of all free cycle intermediates (Fig. 3b). Thus, for the binding of bacitracin to UPP, which constitutes the largest pool of lipid II cycle intermediates, our model predicts only a marginal in vivo efficacy gap, i.e., an $IC_{50}$ very similar to the in vitro $K_D$ value.

Next, we focussed on the commonly used food preservative nisin—a polycyclic antibacterial peptide that binds with high affinity to lipid II, the latter of which constitutes the smallest pool of externally accessible cycle intermediates. To our surprise, for nisin our model predicted an $IC_{50}$ value ($IC_{50}^{NIS} = 10\,\mu M$) about 700-fold higher than the in vitro dissociation constant entering

the model simulation ($K_D^{NIS} = 0.015\,\mu M$[48]) (Fig. 3c)—qualitatively similar to the in vivo efficacy gap reported in literature (see Introduction). What is the origin of this discrepancy in the model? When again considering the relative abundances of cycle intermediates at varying antibiotic concentrations, it turns out that nisin—at low levels around the $K_D$ value—also effectively binds to its target, leading to a pool level of nisin-lipid II complexes comparable to the free form of lipid II (Fig. 3d). However, this sequestration of lipid II only corresponds to ~1% ($10^4$ molecules) of the total number of lipid carriers in the cycle (Fig. 3d, g), thereby not reducing the overall abundance of free carriers significantly. Accordingly, the circular flux of carriers within the lipid II cycle quickly replenishes the free form of lipid II molecules and leads to a similar PG synthesis rate as in the absence of nisin (99% of max). Only when increasing the nisin concentration 700-fold over its $K_D$ value, the amount of sequestered carriers (nisin-lipid II) rises to ~50% of the total abundance of cycle intermediates (Fig. 3d, h), thereby reducing also the free lipid II pool and hence the overall PG synthesis rate to 50% of its maximal value (Fig. 3c, d, h). Thus, within our model the small pool size of externally accessible lipid II (~1/100 of total lipid carriers) leads to inefficient sequestration of lipid carriers, thereby reducing the susceptibility of cell wall biosynthesis towards lipid II-binding antibiotics. In contrast, the binding of bacitracin to external UPP, constituting the largest pool of cycle intermediates (~2/3 of total lipid carriers), leads to efficient sequestration of lipid carriers already at concentrations around the $K_D$ value (Fig. 3e, f). In summary, these results indicate that the in vivo efficacy gap results from asymmetric distributions of externally accessible targets, and that the discrepancy between $K_D$ and $IC_{50}$ increases for decreasing target pool sizes.

To assess the predictive power of our model, we next compared the theoretical $IC_{50}$ values with experimentally determined MICs for the given antibiotics (Fig. 3a and Supplementary Table 4b). On first sight, the in vivo MIC of wildtype *B. subtilis* strain W168 ($MIC^{BAC} = 180\,\mu M$ bacitracin[49]) was ~100-fold higher than the predicted $IC_{50}$ value ($IC_{50}^{BAC} = 1.8\,\mu M$). However, the model did not factor in the action of the BceAB resistance pump, which confers high levels of bacitracin resistance to wildtype *B. subtilis* cells[49]. The MIC of a strain deleted for *bceAB* (W168 Δ*bceAB*; $MIC^{BAC} = 1.7\,\mu M$ bacitracin[49]); instead closely matches the model-predicted $IC_{50}$, confirming that the in vivo efficacy gap is only ~two-fold for the UPP-binding antibiotic bacitracin (Fig. 3a). Similarly, the model prediction for nisin ($IC_{50}^{NIS} = 10\,\mu M$) is only a factor of two higher than the experimental MIC of a strain deleted for the primary nisin resistance determinant (W168 Δ*psdAB*; $MIC^{NIS} = 4.8\,\mu M$[11]), revealing a 330-fold higher in vivo MIC compared to the in vitro $K_D$ (Fig. 3c). Here, the slightly lower experimental $MIC^{NIS}$ compared to the predicted $IC_{50}^{NIS}$ might be caused by membrane pore formation triggered by high nisin levels[16,50,51], which will increase the potency of nisin but is not reflected in the model. These results indicate that our model provides an accurate description of the lipid II cycle under antibiotic treatment, and allows for precise predictions of the in vivo antibiotic susceptibility from first principles.

**Analytical expression of the in vivo efficacy gap.** Next, we wanted to derive an intuitive mathematical formula describing how antibiotic susceptibility depends on the pool size of the targeted lipid carrier, thereby rationalizing the origin of the in vivo efficacy gap. To obtain a closed analytical expression for the PG synthesis rate as a function of the antibiotic concentration, we considered a simplified model of the lipid II cycle (see Methods and Fig. 4a): This model takes into account first order reactions between the antibiotic, $A$, and its unbound lipid carrier

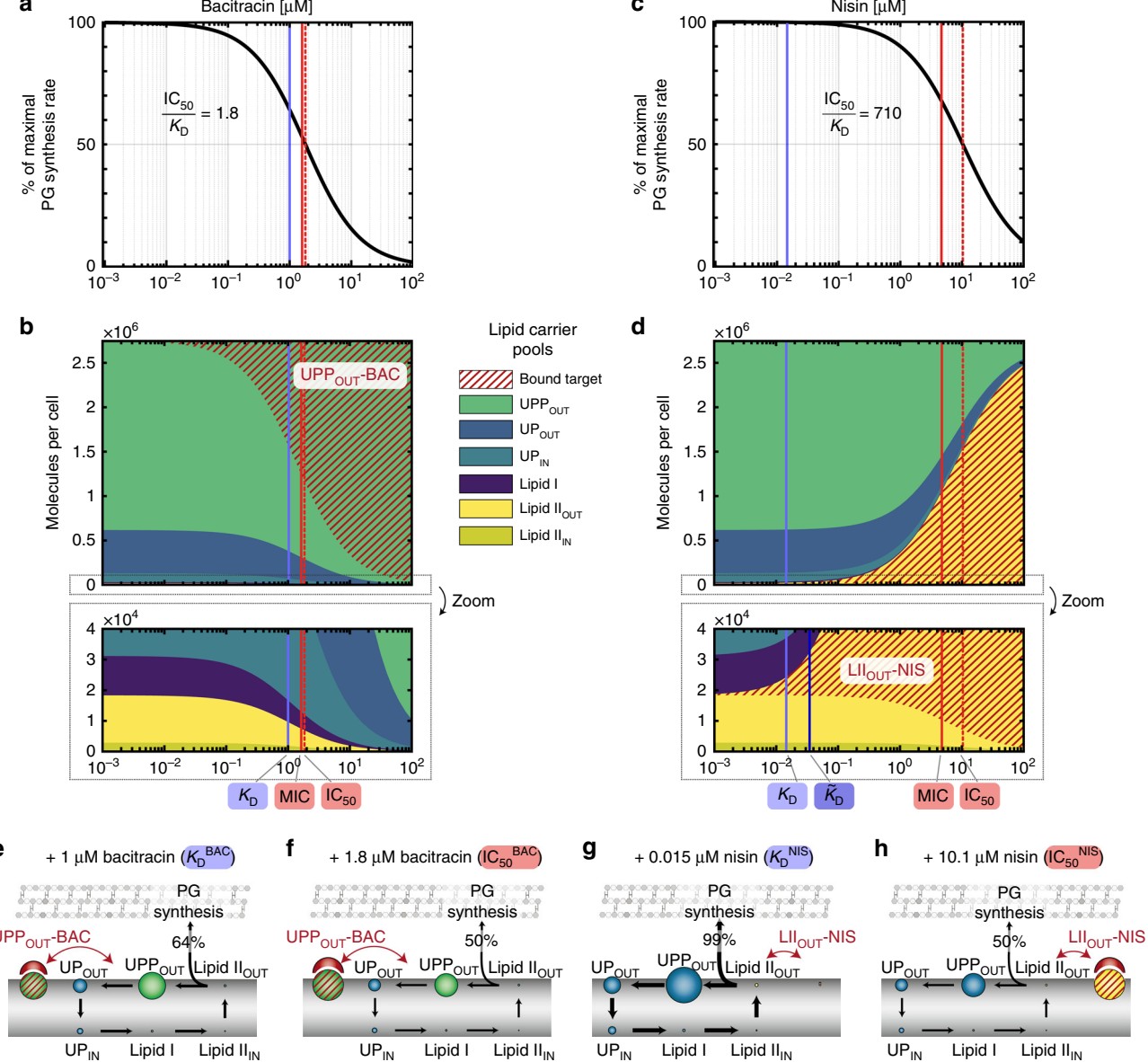

**Fig. 3** The asymmetric distribution of lipid II cycle intermediates can generate a massive in vivo efficacy gap in *B. subtilis*. **a**, **c** Model prediction of the PG synthesis rate (black lines) for varying bacitracin (**a**) and nisin (**c**) concentrations. Blue lines indicate the in vitro dissociation constants $K_D$ and red solid lines the experimental MICs derived from literature (see Supplementary Table 4 for parameter values and references). The red dashed line indicates $IC_{50}$ value predicted by the model, i.e. the concentration at which the PG synthesis reaches 50% of the maximal rate. **b**, **d** Model prediction of the pool levels of lipid II cycle intermediates at different bacitracin (**b**) and nisin (**d**) concentrations. **e**–**h** Schematic illustrations of pool level distributions at bacitracin and nisin concentrations corresponding to the respective $K_D$ and $IC_{50}$ values. **e** At a bacitracin concentration equal to the $K_D$ value, the external UPP pool is significant reduced, implying efficient sequestration of high levels of lipid II cycle intermediates from the cycle, thereby decreasing all free cycle intermediates, especially lipid II. While this already reduces the rate of PG synthesis to a level of 64% of its maximum, only slightly higher bacitracin concentrations (**f**) are required to reduce the rate of PG synthesis to 50%. **g** In contrast, although nisin—at a concentration around the $K_D$—binds 50% of the free lipid II pool, this only sequesters ~1% of all lipid intermediates from the cycle. As the remaining lipid carriers quickly replenish the free form of lipid II molecules by on-going cycling, the rate of PG synthesis is not reduced significantly (99% of maximum). **h** Eventually, high concentrations of nisin are required at the $IC_{50}$ value to sequester ~50% of the total lipid II cycle intermediates into nisin-lipid II complexes, in order to decrease the pool of free lipid II pool and the PG synthesis rate to 50%. For an analysis of the remaining three antibiotics please refer to Supplementary Fig. 3

target, $S_{unbound}$, with association- and dissociation rates $k_{ass}$ and $k_{diss}$, respectively. Moreover, all other (non-target) lipid carrier intermediates of the cycle are summarised as a bactoprenol reservoir, $S_{reservoir}$, which can be interconverted into the unbound lipid carrier with first order rate constants $k_1$ and $k_{-1}$, leading to PG synthesis at a rate $j_{PG} = k_{-1}[S_{unbound}]$ (and in equilibrium also $j_{PG} = k_1[S_{reservoir}]$). Under the assumption that the lipid II cycle runs much faster than the doubling rate ($k_1, k_{-1} \gg \gamma$) the PG

synthesis rate decreases with the antibiotic concentration $[A]$ according to

$$j_{PG} \sim \frac{\tilde{K}_D(1 + \tilde{K}_G)}{[A] + \tilde{K}_D(1 + \tilde{K}_G)}, \qquad (1)$$

highly reminiscent of the hyperbolic decrease observed in the full model above (cf. Fig. 3a, c). Interestingly, the half-maximal rate of PG synthesis is reached at an antibiotic concentration of

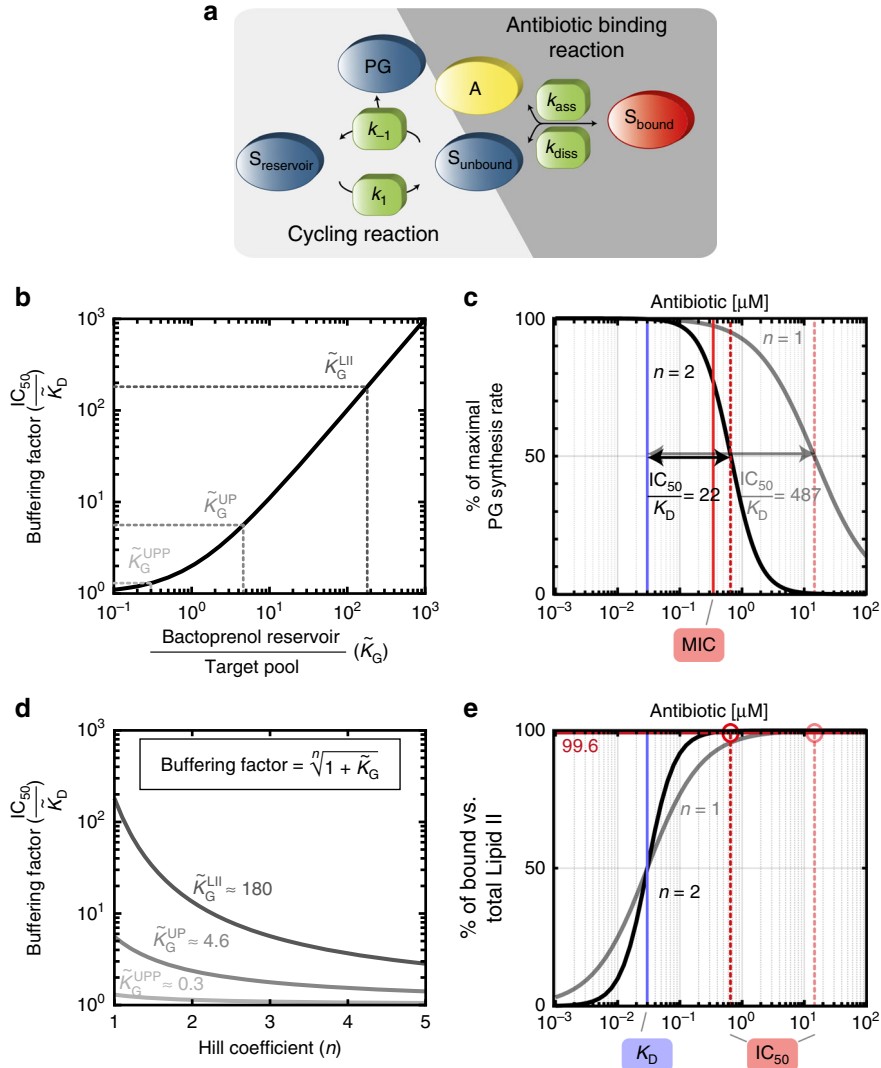

**Fig. 4** A reduced model for the lipid II cycle rationalises the in vivo efficacy gap and elucidates the boost of antibiotic potency by cooperative drug-target interactions. **a** The reduced model for the lipid II cycle considers antibiotic ($A$) binding to its lipid carrier target ($S_{\text{unbound}}$) via first order reactions with association- and dissociation rates $k_{\text{ass}}$ and $k_{\text{diss}}$, respectively, leading to the antibiotic-bound form of the target ($S_{\text{bound}}$). The model summarises all non-target lipid II cycle intermediates as a bactoprenol reservoir ($S_{\text{reservoir}}$), which can be converted into the unbound lipid target intermediate and *vice versa* via first order kinetics at rate constants $k_1$ and $k_{-1}$, respectively. **b** The buffering factor $\left(\frac{\text{IC}_{50}}{K_{\text{D}}} = 1 + \tilde{K}_{\text{G}}\right)$, which is the major determinant of the in vivo efficacy gap, increases for increasing bactoprenol reservoir size relative to the unbound target pool in the absence of the antibiotic ($S_{\text{target}}$) according to $\tilde{K}_{\text{G}} = \frac{S_{\text{reservoir}}}{S_{\text{target}}}$. Buffering factors are indicated for antibiotics binding to external lipid II ($\tilde{K}_{\text{G}}^{\text{LII}}$), external UP ($\tilde{K}_{\text{G}}^{\text{UP}}$), or UPP ($\tilde{K}_{\text{G}}^{\text{UPP}}$). **c** Influence of cooperative drug-target interaction on the IC$_{50}$ value predicted for lipid II-binding antibiotics. Assuming identical in vitro $K_{\text{D}}$ values (corresponding to the in vitro value of vancomycin; blue line) the model predicts that for an antibiotic variant binding in a cooperative manner (Hill coefficient $n = 2$; black and red dashed lines), the IC$_{50}$ is approximately 22 times lower than for a non-cooperative antibiotic binding ($n = 1$; grey and pale red lines). The experimentally measured MIC for vancomycin (*red solid line* [9]), is close to the predicted IC$_{50}$ for the cooperative variant. **d** Scaling of the in vivo efficacy gap with the Hill coefficient $n$ within the reduced model (see Supplementary Fig. 2 for further details). **e** For the simulated vancomycin variants (binding to lipid II), the reduction of the PG synthesis rate to 50% requires that the equilibrium is strongly shifted towards the bound form of lipid II (99.6% bound vs. total lipid II). As increasing Hill coefficients $n$ generally lead to steeper binding curves, the required level of target binding is achieved at a 22-fold lower antibiotic concentration by cooperative antibiotic binding (black) compared to the non-cooperative variant (grey)

IC$_{50} \equiv \tilde{K}_{\text{D}}\left(1 + \tilde{K}_{\text{G}}\right)$, which is strikingly different from the naïve in vitro expectation for the antibiotic-target interaction (IC$_{50} = K_{\text{D}}$). Indeed, the IC$_{50}$ in vivo is not only governed by the biochemical properties of antibiotic binding, but strongly influenced by two factors governing the lipid II cycling reactions: First, the dissociation constant for the antibiotic-target interaction is substituted by the in vivo dissociation constant, $\tilde{K}_{\text{D}} = \frac{k_{\text{diss}} + \gamma}{k_{\text{ass}}}$, which can deviate up to approximately three-fold from the in vitro value $\left(K_{\text{D}} = \frac{k_{\text{diss}}}{k_{\text{ass}}}\right)$, depending on the growth rate $\gamma$, as

well as the kinetics of antibiotic binding and unbinding from its target (see Supplementary Note 1, Supplementary Table 4c and Supplementary Fig. 4c, e). Second, the in vivo dissociation constant is scaled by a buffering factor, which we define as $\left(1 + \tilde{K}_{\text{G}}\right)$. Here $\tilde{K}_{\text{G}} = \frac{[R]}{[T]} \approx \frac{k_{-1}}{k_1}$ describes the ratio between the size of the bactoprenol reservoir (serving as a buffer) and the size of the carrier target in the absence of antibiotic (Fig. 4b). For example, if the buffering reservoir is small compared to the target pool $\left(\tilde{K}_{\text{G}} \ll 1\right)$, the model predicts only a marginal shift in the

$IC_{50}$ $\left(IC_{50} \approx \tilde{K}_D\right)$, indicating that in this case 50% of the total bactoprenol carriers are easily sequestered by an antibiotic concentration equal to the in vivo $\tilde{K}_D$ value. In contrast, if the buffering reservoir is large compared to the target pool $\left(\tilde{K}_G \gg 1\right)$, an antibiotic concentration equal to the in vivo $\tilde{K}_D$ value only sequesters a small amount of the overall bactoprenol carrier level, leading to substantial shifts in the $IC_{50}$ $\left(IC_{50} \gg \tilde{K}_D\right)$. Specifically, when considering that the external lipid II pool in *B. subtilis* (15 μM) is expected to be much smaller than the sum of all other carrier intermediates (2700 μM), the model predicts a buffering factor for the lipid II binding nisin of $\left(1 + \tilde{K}_G\right) \approx$ 180-fold (Fig. 4b). In contrast, for the UPP pool (2100 μM) the other lipid carriers constitute a much smaller reservoir (600 μM), which leads only to a marginal buffering factor for bacitracin of $\left(1 + \tilde{K}_G\right) \approx$ 1.3-fold (Fig. 4b). These results demonstrate that the asymmetric distributions of lipid carrier intermediates lead to a buffering effect against antibiotic attacks, which is particularly pronounced for lipid II binding antibiotics, displaying a several 100-fold in vivo efficacy gap. Thus, although other factors, such as the difference between the in vitro and in vivo dissociation constant and enzyme saturation (see Supplementary Note 1 and Supplementary Fig. 3g) have additional impact on antibiotic susceptibility in the full model, the buffering effect is the major cause for the in vivo efficacy gap for antibiotics targeting small lipid carrier pools.

**Cooperative drug-target interaction boosts antibiotic efficacy.** Next, we focussed on another long-standing debate in the field of antibiotic resistance research, which is related to the effect of cooperative drug-target interaction on antibiotic susceptibility. Here, it is well documented that antibiotics binding in a cooperative manner to lipid II cycle intermediates, e.g., via multimeric complex formation with the target, have a higher potency than antibiotic variants unable to multimerise[23,52–54]. For instance, dimer formation plays a key role in the efficient action of the clinically important antimicrobial peptide vancomycin, as well as in many other glycopeptides[52–54], and has been recognised to enhance the potency of engineered antimicrobial peptides[23]. However, until now the mechanism behind the cooperativity-induced activity boost remained elusive.

We therefore studied the quantitative impact of cooperative drug-target interactions on antibiotic efficacy within our mathematical model. To this end we considered the case of vancomycin, which has a dissociation constant of $K_D^{VAN} =$ 0.03 μM[15] and interacts with lipid II molecules[55] via vancomycin dimerization[56]. In vitro, cooperative antibiotic-target interactions typically lead to sigmoidal binding curves of the form $[A]^n / \left(K_D^n + [A]^n\right)$, with a Hill coefficient $n$ ranging between 1–2 for dimeric binding. For instance, if drug dimerization occurs at a concentration around the dissociation constant to the target, the Hill coefficient will be close to 2, while both strong and weak dimerization relative to target binding will generally lead to $n < 2$ (see Methods and Supplementary Fig. 2c). When analysing the effect of two hypothetical vancomycin variants with identical $K_D$, but different Hill coefficients within our model (Fig. 4c), we find that for a non-cooperatively binding variant ($n = 1$) the model predicts an in vivo efficacy gap similar to nisin $\left(\frac{K_D}{MIC} = 470\right)$, while this is significantly reduced for a cooperatively binding vancomycin variant ($n = 2$), for which the model predicts a 20-fold lower efficacy gap $\left(\frac{K_D}{MIC} = 22\right)$. Interestingly, the experimentally measured MIC for vancomycin in *B. subtilis* ($MIC^{VAN} =$ 0.35 μM[9]) is remarkably similar to the value predicted for the cooperatively binding variant ($IC_{50}^{VAN} = 0.65$ μM) (Fig. 5),

consistent with the observation that dimerization of vancomycin is key for blocking the lipid II pool. Strikingly, also for the dimeric glycolipodepsipeptide ramoplanin, our model predicts almost the same in vivo efficacy gap as for vancomycin ($IC_{50}^{RAM} =$ 0.41 μM), which we find in excellent quantitative agreement with experimental data ($MIC^{RAM} = 0.49$ μM[10]) (Fig. 5).

Why does dimerisation have such a drastic influence on the $IC_{50}$ in our model? To rationalise this behaviour, we extended the simplified model (Fig. 4a) by accommodating cooperative drug-target interactions (see Methods). Under similar assumptions as in the previous section ($k_1, k_{-1} \gg \gamma$) the PG synthesis rate now takes the form

$$j_{PG} \sim \frac{\tilde{K}_D^n\left(1 + \tilde{K}_G\right)}{[A]^n + \tilde{K}_D^n\left(1 + \tilde{K}_G\right)}. \qquad (2)$$

Following a similar rationale as before, the half-maximal rate of PG synthesis is now reached at an antibiotic concentration of $IC_{50} = \tilde{K}_D \sqrt[n]{1 + \tilde{K}_G}$, where the generalised buffering factor $\sqrt[n]{1 + \tilde{K}_G}$ gets attenuated by the Hill coefficient via the $n$-th root. Thus, the higher the cooperativity $n$, the lower the buffering factor and the smaller the gap between MIC and $K_D$ value. This attenuation of the buffering effect is particularly pronounced if the buffering factor is large, e.g. for antibiotics targeting the lipid II pool $\left(\tilde{K}_G \approx 180\right)$ (Fig. 4d). Intuitively, the mitigating role of cooperativity can be understood as follows: Since the buffering factor is large, the cycle is slowed down only if the antibiotic-target complexes vastly exceed the free target, such that the $IC_{50}$ for lipid II-binding antibiotics is only achieved if the ratio between bound and unbound lipid II molecules is 99.6% (Fig. 4e). Clearly, if the drug-target interaction follows a sigmoidal binding kinetics (as incurred by a Hill coefficient $n = 2$), a similar level of target binding is achieved at a 22-fold lower antibiotic concentration compared to hyperbolic binding kinetics ($n = 1$) (Fig. 4e). This explains why in vivo vancomycin ($n = 2$) is drastically more active than nisin ($n = 1$), although both antibiotics have almost identical in vitro dissociation constants to lipid II. Thus, cooperativity in drug-target interactions can greatly boost the vivo efficacy of the drug by more efficiently sequestering the target as soon as the $K_D$ value is exceeded.

However, we also noted that cooperative drug-target interactions do not always confer such drastic effects. For antibiotics targeting the largest pools of cycle intermediates, UPP and UP, the respective buffering factors are already low ($1 + \tilde{K}_G \approx 1.3$ and $1 + \tilde{K}_G \approx 5.6$, respectively) such that increasing cooperativity only leads to a mild reduction of the buffering effects in our model, with virtually no change for UPP-binding antibiotics (Fig. 4d). Only for UP-binding antibiotics the model predicts that changing cooperativity from monomeric ($n = 1$) to dimeric target binding ($n = 2$) will increase antibiotic potency by a factor $\sqrt{5.6} \approx 2.4$. Interestingly, while it has been controversial whether the UP-targeting lipopeptide antibiotic friulimicin binds its target as monomer or dimer[57], our model predictions for monomeric binding ($IC_{50}^{FRI} = 1.46$ μM) are in excellent agreement with the experimental susceptibility in *B. subtilis* ($MIC^{FRI} = 1.15$ μM[58]) (Fig. 5), suggesting that friulimicin inhibits its target in a non-cooperative manner.

**Discussion**

Over the last decade quantitative experimentation and theoretical modelling has fostered significant progress in our understanding of antibiotic action against bacteria[3,59–61]. While previous theory uncovered a range of non-trivial effects in the action of ribosome-targeting antibiotics[1], our work rationalises similarly

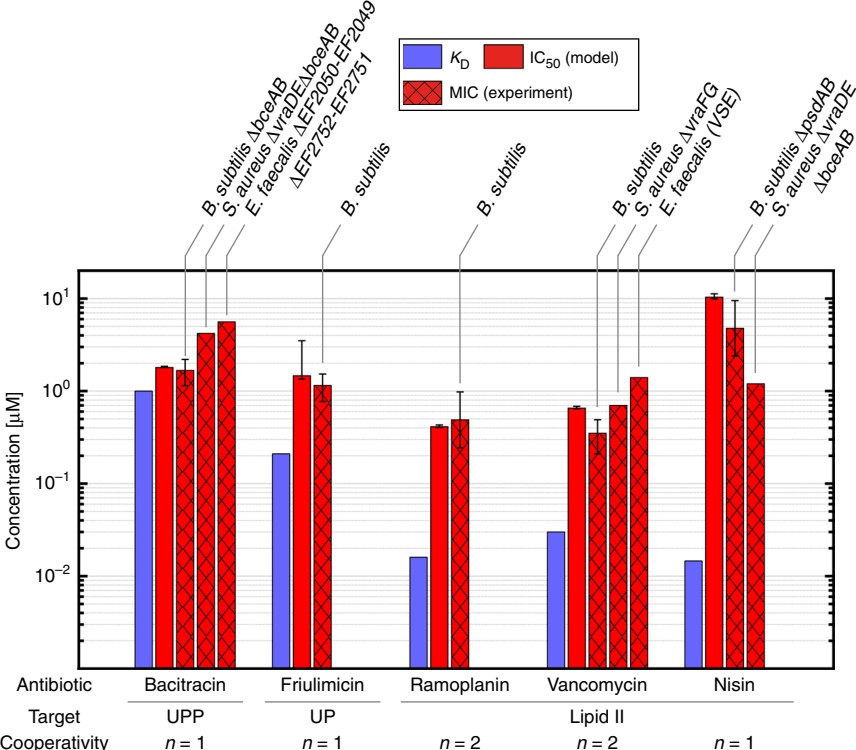

**Fig. 5** Prediction of in vivo efficacy for various cell wall antibiotics in diverse Gram-positive organisms. The mathematical model predicts the in vivo efficacy ($IC_{50}$, red solid bars) exclusively from the antibiotics in vitro dissociation constants ($K_D$, blue bars) and available information about the cooperativity in antibiotic-target-interaction. The model predictions are in good agreement with experimental data published for *B. subtilis*[9–11,49,58], *S. aureus*[12,13] and *E. faecalis*[14,79] strains deleted for the known resistance determinants against the different antibiotics (MIC, red hashed bars), highlighting the universality of the theoretical model for various Gram-positive organisms. Error bars of literature MIC values either represent standard deviations from multiple measurements, when available in the original publications, or were derived from the increments of the serial dilution steps in the published MIC assays. Error bars for the model predictions of $IC_{50}$ values represent confidence intervals propagated from uncertainties in model parameters (see Supplementary Fig. 4 and[80])

counterintuitive effects for cell wall antibiotics. In particular, our theory predicts an inverse correlation between the tolerance towards substrate-sequestering cell wall antibiotics and the abundance of their cellular target, suggesting the principle of minimal target exposure as an intrinsic resistance mechanism against cell wall antibiotics. We show that resistance emerges from the cyclic nature of the cell wall biosynthetic pathway, in which high-abundance intermediates provide a buffer against sequestration of low-abundance intermediates. In this light it seems plausible that bacteria may have evolved to minimise the abundance of externally exposed lipid II molecules, e.g. by speeding up the rate of PG monomer insertion into the cell wall, in order to evade blocking by lipid II-binding antibiotics, which are ubiquitously produced by competing species, such as *Lactobacillus lactis* (nisin), *Amycolatopsis orientalis* (vancomycin) or *Actinomycetes* species (ramoplanin)[62–65].

Our theory further resolves a longstanding conundrum dating back to the 1990s, where it was first observed that cooperative drug-target interactions play a crucial role in the in vivo efficacy of glycopeptide antibiotics[18,22]. Although molecular studies have revealed how cooperativity can emerge from glycopeptide–lipid II interactions[66], it remained enigmatic why it has such a drastic effect on antibiotic efficacy in vivo. Our work reveals that cooperativity alleviates the buffering effect within the lipid II cycle, such that much less antibiotic is required to achieve a similar level of target-inhibition compared to a non-cooperatively binding drug (Fig. 4e). Interestingly, our results indicate that the most pronounced advantage of cooperative drug-target binding arises when the buffering effect is large (Fig. 4d), and to our knowledge,

all cooperatively acting antibiotics bind to lipid II, for which this is the case.

Taken together, our theory correctly predicts the in vivo action of five different antibiotics against the Gram-positive model organism *B. subtilis* (Fig. 5). Since our theory is based on cumulative information about lipid II cycle properties in diverse bacterial species, we wondered whether the derived principles also apply to other organisms, including clinically relevant bacteria. Indeed, *S. aureus* strains deprived of all known resistance determinants also display pronounced in vivo efficacy gaps for nisin ($MIC^{NIS}/K_D^{NIS} = 82$ [13]) and for vancomycin ($MIC^{VAN}/K_D^{VAN} = 24$ [12]), and also the MIC of *E. faecalis* against vancomycin exceeds the in vitro dissociation constant 47-fold[14]—all very similar to the values in *B. subtilis* (Fig. 5 and Supplementary Table 4b). Thus, both the in vivo efficacy gap as well as its molecular origin—namely the asymmetric distribution of lipid II cycle intermediates—seem to be conserved between diverse Gram-positive organisms, highlighting the universality of our model.

Besides the prediction of antibiotic susceptibility in vivo, our theory provides clues about physiological features of the lipid II cycle, which have not been experimentally accessible to date. For instance, our quantitative considerations of lipid II cycling put constrains on the enzyme copy numbers, their reaction kinetics and, most notably, on the so far unknown processes of UP and UPP flipping. Here, our analysis indicates that the flipping of UPP from the inner to the outer leaflet of the cytoplasmic membrane is about as fast as lipid II flipping, both of which are >100-fold faster than the flipping of UP from the outer to the

inner leaflet (Supplementary Table 2). Given that the dual negative charge of the UPP headgroup energetically strongly disfavours spontaneous flip-flop between bilayers[67], such rapid flipping can only be achieved by active UPP transport across the membrane, but a specific UPP flippase has yet to be discovered[68,69]. The 500-fold slower UP flipping suggests that it might follow a passive flip-flop mechanism driven by concentration- and/or charge-gradients, but further experiments are needed to shed light on this.

Another intriguing insight from literature mining was that Gram-negative and -positive bacteria feature similar levels of lipid II cycle-associated enzymes per surface area (Fig. 2b), despite their vastly different demand for peptidoglycan synthesis (Supplementary Table 3a, b). As a consequence, our analysis suggests that the Gram-positive lipid II cycle is driven by faster enzymes, which sacrificed some of their substrate recognition in a speed-affinity trade-off (Fig. 2c). This is consistent with the idea that the increased levels of lipid carrier intermediates found in Gram-positive bacteria are required to saturate these faster enzymes. But why do Gram-positive bacteria not simply produce higher levels of lipid II cycle-associated enzymes to meet this demand? One reason could be that these enzymes are either integral membrane or membrane-associated proteins, such that raising the abundance of the PG synthetic machinery could exceed the carrying capacity of the membrane. Indeed, in *E. coli* the cytoplasmic membrane bears a total of ~33.000 proteins per μm² [70,71], and the sum of all enzymes in the lipid II cycle constitutes ~1–3% of the membrane proteome. Thus, raising the PG synthetic machinery by a factor of 13 to meet the PG demand of Gram-positive bacteria could clearly lead to fitness trade-offs with other essential transport- and biosynthetic processes. Comparative experimental studies of the PG synthetic machinery in Gram-positive and -negative organisms will help to further elucidate the quantitative differences in this rate-limiting step of bacterial cell wall synthesis.

The insights gained here can help guiding the design of new drugs—by suggesting that novel cell wall antibiotics will perturb the lipid II cycle most effectively by (i) binding low-abundant cycle intermediates in a highly cooperative manner or by (ii) targeting the high-abundant intermediate pools. In addition, our model of the lipid II cycle provides the basis for broader analyses of various further classes of cell wall antibiotics, such as drugs inhibiting the enzymes in the lipid II cycle (e.g. beta-lactams inhibiting PBPs) or drugs targeting the substrates of PG precursor production (e.g. fosfomycin). To this end, the model will need to be expanded, e.g., to explicitly incorporate the biochemical characteristics and copy numbers of all redundant PBPs (as opposed to treating them as one effective reaction, as in the present model), highlighting the importance of further biochemical studies of PBPs and other lipid II cycle-associated enzymes for developing a complete systems-level description of this essential cellular pathway. Likewise, the seamless biochemical characterization of enzymes involved in PG precursor synthesis and cell wall recycling will enable quantitative modelling of drugs interfering with these important aspects of cell wall synthesis. Hence, the presented model serves as an excellent starting point to develop a whole-cell model of antibiotic action. One important aspect will be the development of theoretical models describing the regulation and action of known resistance mechanisms (which are deleted in the strains considered in this work), to provide a systems-level description of antibiotic action in wild-type cells. First steps in developing such models have been made, e.g., for the bacitracin resistance determinant BceAB in *B. subtilis*[72] and for beta-lactamases in *S. aureus*[73]. Coupling our theory of wall synthesis with the bacterial growth laws[1,4–6] will lead to new insights into the growth-rate dependency of antibiotic action

and may advance our understanding of antibiotic tolerance of slow- and non-growing cells[74]. Beyond this, a comprehensive model will contribute to a quantitative understanding of whole-cell physiology, which is the starting point to predict drug–drug interactions between antibiotics targeting different physiological pathways. Finally, we believe that such rational approaches to understand the physiological targets of antibiotics are urgently needed to develop novel strategies in our fight against antimicrobial resistance.

## Methods

**Mathematical model of the lipid II cycle**. Our computational model of cell wall synthesis focuses on the core reactions of the lipid II cycle and describes peptidoglycan synthesis for each individual cell (Supplementary Fig. 1c). Time-dependent changes of the pool levels of lipid II cycle intermediates are described by deterministic differential equations to monitor the dynamics of cell wall synthesis. Diverse model assumptions, based on the current state of knowledge about the lipid II cycle, determine the frame of the kinetic model:

(i) The individual states of lipid carrier are included as time-dependent variables in the model, distinguishing between lipid carriers localised in the inner (IN) and outer (OUT) leaflet of the cytoplasmic membrane:

- $UPP_{IN}$ = internal pool of undecaprenyl pyrophosphate (UPP)
- $UPP_{OUT}$ = external pool of undecaprenyl pyrophosphate
- $UP_{IN}$ = internal pool of undecaprenyl phosphate (UP)
- $UP_{OUT}$ = external pool of undecaprenyl phosphate
- $LI$ = pool of lipid I
- $LII_{IN}$ = internal pool of lipid II
- $LII_{OUT}$ = external pool of lipid II

(ii) The cytoplasmic production of soluble PG precursors (UDP-MurNAc-pentapeptide and UDP-GlcNAc) is not described in detail in the model. Since the precursor pool levels are homeostatically controlled[34,35] at sufficiently high levels to saturate the enzymes of the corresponding reactions (Supplementary Table 1), the rate of cell wall synthesis is normally not limited by PG precursor abundance. Although we are well aware that this assumption does not accurately reflect the situation when PG precursor synthesis itself is targeted, e.g. by fosfomycin, it is plausible to assume constant pools of PG precursors when considering antibiotics targeting the membrane-anchored steps of PG synthesis only.

(iii) The de novo synthesis of UPP in the cytoplasm has to balance the overall growth-driven dilution of all lipid II cycle intermediates. To this end, we assume a constant UPP production rate $\alpha = \frac{\ln(2)}{T_D} \sum [S_i]$ in the cytoplasm, balancing dilution within one generation time $T_D$. Likewise, the growth-dependent dilution of all individual lipid intermediate pools occurs at a rate $\gamma = \frac{\ln(2)}{T_D}$.

(iv) The individual enzymatic reactions are modelled by Michaelis-Menten kinetics, for which substrate levels ($S_i$), enzyme levels ($E$), catalytic constants of the enzymes ($k_{cat}$) as well as the Michaelis–Menten constants ($K_M$) parameterise the reaction dynamics.

(v) Since the biochemical properties of the enzymes catalysing the flipping reaction of lipid II ($LII_{IN}$ to $LII_{OUT}$) are largely unknown, and the flipping of UPP ($UPP_{IN}$ to $UPP_{OUT}$) and UP ($UP_{OUT}$ to $UP_{IN}$) was only hypothesised, for parsimony reasons we assumed first order kinetics for these reactions, as quantified by an effective rate constant $k_i$ (i = UP, UPP, LII).

Under these assumptions the following set of ordinary differential equations describes the time-dependent changes of the lipid II cycle intermediate pools and the concomitant effect on the rate of PG synthesis, $j_{PG}$

$$\frac{d[UPP_{IN}]}{dt} = \alpha - k_{UPP}[UPP_{IN}] - \gamma[UPP_{IN}] \tag{3}$$

$$\frac{d[UPP_{OUT}]}{dt} = k_{UPP}[UPP_{IN}] - v_{max}^{UppPs} \frac{[UPP_{OUT}]}{K_M^{UppPs} + [UPP_{OUT}]} + v_{max}^{PBPs} \frac{[LII_{OUT}]}{K_M^{PBPs} + [LII_{OUT}]} - \gamma[UPP_{OUT}] \tag{4}$$

$$\frac{d[UP_{OUT}]}{dt} = v_{max}^{UppPs} \frac{[UPP_{OUT}]}{K_M^{UppPs} + [UPP_{OUT}]} - k_{UP}[UP_{OUT}] - \gamma[UP_{OUT}] \tag{5}$$

$$\frac{d[UP_{IN}]}{dt} = k_{UP}[UP_{OUT}] - v_{max}^{MraY} \frac{[UP_{IN}]}{K_M^{MraY} + [UP_{IN}]} - \gamma[UP_{IN}] \tag{6}$$

$$\frac{d[LI]}{dt} = v_{max}^{MraY} \frac{[UP_{IN}]}{K_M^{MraY} + [UP_{IN}]} - v_{max}^{MurG} \frac{[LI]}{K_M^{MurG} + [LI]} - \gamma[LI] \tag{7}$$

$$\frac{d[\mathrm{LII_{IN}}]}{dt} = v_{\max}^{\mathrm{MurG}} \frac{[\mathrm{LI}]}{K_{\mathrm{M}}^{\mathrm{MurG}} + [\mathrm{LI}]} - k_{\mathrm{LII}}[\mathrm{LII_{IN}}] - \gamma[\mathrm{LII_{IN}}] \tag{8}$$

$$\frac{d[\mathrm{LII_{OUT}}]}{dt} = k_{\mathrm{LII}}[\mathrm{LII_{IN}}] - v_{\max}^{\mathrm{PBPs}} \frac{[\mathrm{LII_{OUT}}]}{K_{\mathrm{M}}^{\mathrm{PBPs}} + [\mathrm{LII_{OUT}}]} - \gamma[\mathrm{LII_{OUT}}] \tag{9}$$

$$j_{\mathrm{PG}} = v_{\max}^{\mathrm{PBPs}} \frac{[\mathrm{LII_{OUT}}]}{K_{\mathrm{M}}^{\mathrm{PBPs}} + [\mathrm{LII_{OUT}}]} \tag{10}$$

**Simulations of antibiotic treatment.** In order to accommodate cell wall antibiotic treatment in the theoretical model of the lipid II cycle, we considered ligand-binding between the antibiotic ($A$) and its target ($T$) (where T can be any of the dynamic variables in Eqs. (3–9))

$$[A] + [T] \overset{K_D}{\longleftrightarrow} [AT],$$

with the in vitro equilibrium dissociation constant $K_D = \frac{k_{\mathrm{diss}}}{k_{\mathrm{ass}}}$, defined as the ratio between dissociation and association rate, respectively. Consequently, the model for the lipid II cycle defined in Eqs. (3–10) was extended by one differential equation describing the dynamics of the antibiotic-bound lipid intermediate pool ($AT$),

$$\frac{d[AT]}{dt} = k_{\mathrm{ass}}[A][T] - k_{\mathrm{diss}}[AT] - \gamma[AT]. \tag{11}$$

Since the individual dissociation and association rates were rarely studied in vitro, we set the association rate to the fixed value of $k_{\mathrm{ass}} = 0.75\,\mu\mathrm{M}^{-1} \times \mathrm{min}^{-1}$ (as measured for the binding of bacitracin to its target UPP [72]) and calculated the dissociation rates from experimentally determined in vitro dissociation constants $K_D = \frac{k_{\mathrm{diss}}}{k_{\mathrm{ass}}}$ (Supplementary Table 4a). As we are well aware that association rates can be different for different antibiotics, we subsequently investigated the robustness of our model predictions against variations in the association rates (see Supplementary Note 1; Influence of binding dynamics on the $\mathrm{IC_{50}}$).

Given that the five antibiotics analysed here vary in both the binding dynamics (quantified by the $K_D$ values) as well as the cooperativity of antibiotic-target-interactions (defined by the Hill coefficient $n$) (Supplementary Table 4a), we integrated an effective quantitative description of the multimer formation as well as the antibiotic binding reaction into our model:

$$n[A] \overset{K_{\mathrm{coop}}}{\longleftrightarrow} [A_n] + [T] \overset{K_D}{\longleftrightarrow} [A_n T]$$

$$K_{\mathrm{eff}} = K_{\mathrm{coop}} \times K_D.$$

Since it was not always clear (e.g. in case of vancomycin) whether the antibiotic multimerisation occurs before or after target-binding, and also the stoichiometry within the antibiotic-target-complex was not always known precisely, we asked if all differential binding scenarios generate cooperativity, i.e. a Hill coefficient $n > 1$. To this end, we deduced the Hill expression describing the probability of bound and thereby inactivated target $P_{\mathrm{bound}}$ from analysing all possible states of antibiotic-target-interaction (Supplementary Fig. 2a, b) and estimated the Hill coefficient $n$ arising from this (Supplementary Fig. 2c). Here, the Hill coefficient $n$ and thereby the cooperativity reaches its maximum if the dissociation constants of multimer formation and antibiotic binding are comparable, i.e. $K_{\mathrm{coop}} \approx K_D$. Obviously, if one of the two reactions dominates the other, that is dimerization is significantly weaker than target binding or vice versa, the effect of cooperativity disappears. Hence, in order to study cooperativity in our model, we took a coarse-grained approach assuming an effective Hill coefficient and binding threshold $K_{\mathrm{eff}}^n = \frac{k_{\mathrm{diss}}}{k_{\mathrm{ass}}}$, leading to the following kinetic equation

$$\frac{d[A_n T]}{dt} = k_{\mathrm{ass}}[A]^n[T] - k_{\mathrm{diss}}[A_n T] - \gamma[A_n T] \tag{12}$$

Within this expanded model including the quantitative description of antibiotic-target-interaction, we studied the effect of antibiotic action on the lipid II cycle for the five different cell wall antibiotics. In particular, we determined the antibiotic concentration necessary to decrease the PG synthesis rate to its half-maximal level to quantify the antibiotic efficacy and defined this concentration as the $\mathrm{IC_{50}}$ (Fig. 3a, c and Supplementary Fig. 3a, c, e). Additionally, we analysed the changes in the pool sizes of the different lipid II cycle intermediates $S_i$ under varying antibiotic concentrations (Fig. 3b, d and Supplementary Fig. 3b, d, f).

**Reduced model of the lipid II cycle.** To arrive at an analytical expression for the PG synthesis rate in dependence on the antibiotic concentration, we developed a reduced model of the lipid II cycle (Fig. 4a). Similar to the full model in Eqs. (3–12), the antibiotic ($A$) can bind to its free target ($S_{\mathrm{unbound}}$) within the lipid II cycle with in vitro dissociation constant $K_D = \frac{k_{\mathrm{diss}}}{k_{\mathrm{ass}}}$, leading to a pool of bound target ($S_{\mathrm{bound}}$), but now the sum of all other, non-target lipid II cycle intermediates are represented as 'bactoprenol reservoir' ($S_{\mathrm{reservoir}}$). For simplifying reasons, the inter-conversion of one species into the other follows first order kinetics, determined by

the equilibrium constant $K_G = \frac{k_{-1}}{k_1}$. As in the full model, production of new lipid carriers at rate $\alpha$ balances the overall growth-driven dilution of all reaction species with rate $\gamma$:

$$\frac{d[S_{\mathrm{reservoir}}]}{dt} = \alpha - k_1[S_{\mathrm{reservoir}}] + k_{-1}[S_{\mathrm{unbound}}] - \gamma[S_{\mathrm{reservoir}}] \tag{13}$$

$$\frac{d[S_{\mathrm{unbound}}]}{dt} = k_1[S_{\mathrm{reservoir}}] - k_{-1}[S_{\mathrm{unbound}}] - k_{\mathrm{ass}}[S_{\mathrm{unbound}}][A] + k_{\mathrm{diss}}[S_{\mathrm{bound}}] - \gamma[S_{\mathrm{unbound}}] \tag{14}$$

$$\frac{d[S_{\mathrm{bound}}]}{dt} = k_{\mathrm{ass}}[S_{\mathrm{unbound}}][A] - k_{\mathrm{diss}}[S_{\mathrm{bound}}] - \gamma[S_{\mathrm{bound}}] \tag{15}$$

Here, we assume that the production of new lipid carriers enriches the bactoprenol reservoir ($S_{\mathrm{reservoir}}$), and later consider the scenario in which new lipid carriers feed the free target pool ($S_{\mathrm{unbound}}$), the latter of which is the case for UPP-binding antibiotics. In flux-balance $\left(\frac{d}{dt} = 0\right)$ the fraction of antibiotic-bound target relative to the total abundance of cycle intermediates ($S_{\mathrm{TOT}} = S_{\mathrm{reservoir}} + S_{\mathrm{unbound}} + S_{\mathrm{bound}}$) is given by

$$\frac{[S_{\mathrm{bound}}]}{S_{\mathrm{TOT}}} = \frac{[A]}{[A] + \frac{\gamma}{k_1} + \tilde{K}_D\left(1 + \tilde{K}_G\right)},$$

where $\tilde{K}_D = \frac{k_{\mathrm{diss}} + \gamma}{k_{\mathrm{ass}}}$ and $\tilde{K}_G = \frac{k_{-1} + \gamma}{k_1}$ are the respective in vivo equilibrium constants. Moreover, when new lipid carriers feed the free target pool, arrive at an analytical solution of a similar form

$$\frac{[S_{\mathrm{bound}}]}{S_{\mathrm{TOT}}} = \frac{[A]}{[A] + \tilde{K}_D\left(1 + \tilde{K}_G\right)},$$

where $\tilde{K}_G = \frac{k_{-1}}{k_1 + \gamma}$. However, as the cycling reactions dominate the de novo synthesis, i.e. $k_{-1}, k_1 \gg \gamma$ (see Supplementary Note 1; Quantitative considerations of the peptidoglycan synthesis in E. coli), both solutions can be approximated by

$$\frac{[S_{\mathrm{bound}}]}{S_{\mathrm{TOT}}} \approx \frac{[A]}{[A] + \tilde{K}_D\left(1 + \tilde{K}_G\right)},$$

with $\tilde{K}_D = \frac{k_{\mathrm{diss}} + \gamma}{k_{\mathrm{ass}}}$ and $\tilde{K}_G \approx \frac{k_{-1}}{k_1}$. Likewise, the unbound form of the target takes the form

$$\frac{[S_{\mathrm{unbound}}]}{S_{\mathrm{TOT}}} \approx \frac{\tilde{K}_D}{[A] + \tilde{K}_D\left(1 + \tilde{K}_G\right)},$$

such that the relative reduction of the PG synthesis rate in presence of the antibiotic (concentration $[A]$) compared to the unperturbed synthesis rate is

$$\frac{j_{\mathrm{PG}}([A])}{j_{\mathrm{PG}}([A] = 0)} = \frac{k_{-1}[S_{\mathrm{unbound}}]_{[A]}}{k_{-1}[S_{\mathrm{unbound}}]_{[A]=0}} = \frac{\tilde{K}_D\left(1 + \tilde{K}_G\right)}{[A] + \tilde{K}_D\left(1 + \tilde{K}_G\right)}.$$

When incorporating cooperative drug-target binding as in Eq. (12), we analogously obtain

$$\frac{j_{\mathrm{PG}}([A])}{j_{\mathrm{PG}}([A] = 0)} = \frac{\tilde{K}_D^n\left(1 + \tilde{K}_G\right)}{[A]^n + \tilde{K}_D^n\left(1 + \tilde{K}_G\right)},$$

representing the key result for in vivo antibiotic action in the main text. Thus, the half-maximal rate $j_{\mathrm{PG}}$ is reached at an antibiotic concentration $[A] = \mathrm{IC_{50}} = \tilde{K}_D \sqrt[n]{1 + \tilde{K}_G}$. Given that in the absence of the antibiotic $\tilde{K}_G \approx \frac{k_{-1}}{k_1} \approx \frac{[S_{\mathrm{reservoir}}]}{[S_{\mathrm{target}}]}$, the $\mathrm{IC_{50}}$ clearly scales with the $n$-th root of the ratio between bactoprenol reservoir and target pool in the absence of antibiotic ($S_{\mathrm{target}}$).

Finally, taking cycling rates and pool level distributions equal to the full model into account, we show that the reduced model reproduces the model predictions of the full model (Supplementary Fig. 3g). The only subtle differences arise from the fact that the reduced model considers first order kinetics, leading to a linear dependency between the lipid pool sizes and the individual fluxes from one intermediate to the next. Consequently, a reduction of the pool sizes to 50% of their maxima by antibiotic binding directly leads to a half-maximal rate of PG synthesis. In contrast, the Michaelis-Menten kinetics implemented in the full model features saturation effects. Since the pool levels of lipid carrier are on the same order as the $K_M$ values of the respective enzymes (Supplementary Table 1), most enzymes are on the brink of saturation, indicating that there is not necessarily linear dependency between the flux from substrate to product pool and substrate levels. Indeed, the substrate pools have to be reduced by slightly more than 50% to concomitantly reach a halved PG synthesis rate, requiring slightly higher antibiotic concentrations as predicted from the simplified scenario (Supplementary Fig. 3g).

**Model simulations and parameter fitting.** The numerical solution of the differential equations and all simulations were performed with custom scripts developed in MATLAB$^{\mathrm{TM}}$ R2017b software (The MathWorks, Inc.). To constrain the model to a physiological parameter regime we followed the rationale detailed in the Supplementary Note 1. These constraints lead to eleven objective functions

with seven unknown parameters. To solve this over-determined non-linear data-fitting problem, we used the function *lsqnonlin* imbedded in the MATLAB™ software, solving nonlinear least-squares curve fitting problems of the form

$$min\|f(x)\|_2^2 = min(f_1(x)^2 + f_2(x)^2 + \dots + f_n(x)^2).$$

by using a trust-region-reflective Newton algorithm. As outputs, it returns the optimum $\bar{x}$ of the problem as well as the squared 2-norm $\chi^2$ of the residual at $\bar{x}(\chi^2 = \sum f(\bar{x})^2)$. To account for the presence of local optima, 100 independent fits were performed with randomly chosen initial parameter sets. Eventually, the best-fit result (minimal $\chi^2$) was defined as the final parameter set (Supplementary Table 2).

**Reporting summary**. Further information on research design is available in the Nature Research Reporting Summary linked to this article.

## Data availability
The authors declare that the data supporting the findings of this study are available within the paper and its supplementary information files.

## Code availability
Matlab code used in this project for data analysis is available from the corresponding author upon reasonable request.

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

## Acknowledgements

We thank Susanne Gebhard and Peter Graumann for discussion and suggestions. The research was supported via the Cusanuswerk scholarship programme (Germany), a grant from the Deutsche Forschungsgemeinschaft (DFG, Germany; grant FR3673/1-2) and the LOEWE programme of the State of Hesse (Germany).

## Author contributions

H.P. and G.F. designed research. H.P. performed research and analysed data with input from A.D. and G.F. All authors wrote the manuscript.

## Additional information

**Competing interests:** The authors declare no competing interests.

