## [Peer Review File · Nature Communications]

Reviewers' Comments:

Reviewer #1:

Remarks to the Author:

Piepenbreier et al. present a combination of theoretical and experimental data to explain observed discrepancies between the in vitro and in vivo efficacy of several antibiotics. The authors focus on clinically-important antibiotics that bind to precursors of the essential cell wall that surrounds nearly all bacteria and is composed of peptidoglycan (PG). The authors use published data on reaction rates and precursor concentrations to establish their model in Gram-negative (*E. coli*) and then Gram-positive (*B. subtilis* and pathogens like *S. aureus*) bacteria. The authors present experimental data demonstrating that an antibiotic that inhibits early stages in precursor synthesis (lipid carrier) shows better agreement between in vitro and in vivo efficacy than an antibiotic that targets later stages (the PG monomer Lipid II) due to the "circular flux" nature of PG biosynthesis. Lastly, they illustrate that incorporating cooperativity factors for several drug-target interactions into mathematical models leads to successful predictions of in vivo drug activity, rather than simply using hyperbolic binding models for all antibiotics.

Overall, this work is clearly written, the models are explained with sufficient detail for biochemists/microbiologists, and the conclusions drawn from these data are significant, as they explain discrepancies between in vitro data and in vivo observations that are the subject of much debate for this class of antibiotics. However, some parameters of the model (and some of the discussion text) overlook key biological processes involved in PG synthesis and inhibition across different organisms. The manuscript would be significantly strengthened if the following major points were addressed:

1. In the introduction, the authors state "By incorporating all existing experimental evidence in the literature, our theory accounts for all biochemical knowledge of this essential pathway and reconciles it with the in vivo inhibition patterns under antibiotic treatment." (pg 3, lines 73-75). This statement overreaches the scope of this work, and should be adjusted. The authors only test antibiotics that bind peptidoglycan precursors, and not antibiotics that target peptidoglycan biosynthetic enzymes (such as the PBPs). For example, the beta-lactams, which bind the PBPs, are a very important class of cell wall antibiotics but are not discussed in this paper. This should be clarified in the introduction. Other comments below only re-iterate this point about the true generality of this work.

2. In line with the comment above, it would be helpful if the authors explained if their model could be used to describe the behavior of drugs such as the beta-lactams (covalent inhibitors) and moenomycins (non-covalent inhibitors), which target the PBPs (PG biosynthetic enzymes). For example, does the effect of PBP inhibition on PG monomer (Lipid II substrate) levels have a similar effect to drugs that bind Lipid II (like nisin)?

3. The authors compare the dimensions of the peptidoglycan layer of Gram-negative and -positive bacteria in order to create a mathematical model for PG synthesis for Gram-positives based on the model built for Gram-negatives. The authors state that Gram-positives must require an increased rate of PG synthesis because their cell walls are thicker than that of Gram-negatives but the cells are similarly sized (page 8, lines 191-197). However, a potential role for PG hydrolases in differences between Gram-positive and -negative cell wall thickness is never raised. The introduction and discussion sections, as well as the derivation of the model, would be strengthened if the possible effect of PG degradation enzymes were acknowledged.

4. A defining difference between Gram-negatives and -positives is the presence of an outer membrane in the former. The authors note that one possible (and popular) explanation for discrepancies between

in vitro and in vivo efficacies is "sequestration" (line 61, page 3) or a change in effective concentration in vivo. To broaden the scope of this work, the authors could discuss if their model predicts changes in drug activity measured for Gram-negatives with and without permeabilized outer membranes to explore the validity of the "sequestration" effect. An outer membrane deficient *E. coli* strain (see Eggert et al., *Science*, 2001, 294, 361) may be worthwhile to test experimentally following Figure 1.

5. In line with the comment above, why are only antibiotics typically used to treat Gram-positives tested in this paper? This should be explained in the introduction and/or discussion when relevant (especially when making broad conclusions from this work).

6. The authors state that they "integrate all known biochemical properties" (page 7, line 184) to derive their model; however, this statement is not convincing based on the information provided. For example, in Supplementary Table 2, footnote b states "k_{cat} for the bifunctional transglycosylase PBP1b (only available k_{cat})". In the text, the authors note that characterization of Gram-positive PBPs is "sparse" (page 7, lines 186-187). There have been many kinetic studies done on PBPs from various organisms. For a quick review, please see Sauvage, E. et al. (*FEMS Microbiol Rev.* 2008, 32, 234) Table 1, which contains kinetic parameters for six PBPs (from a mixture of Gram-negative and Gram-positive organisms). Most significantly, PBP1a is listed, which is considered as a major bifunctional PG synthetic enzyme (along with PBP1b) in *E. coli*, and is important to consider in any unified PG model. If the authors excluded this data for a particular reason, it should be explained.

7. In Figure 2b and Supplementary Figure 3, how were Lipid II cycle intermediate concentrations measured with changing antibiotic concentration? Supplementary Table 1 references past reports for levels of intermediates in *E. coli*; were the same techniques employed?

8. Please include a procedure for determining MIC values in the supplemental (or list references in a separate section of the supplemental).

9. The authors suggest at the end of the discussion that this work may help combat resistance. This work clearly describes the ability of cells to compensate for inhibition of substrates by adjusting pathway flux and takes into account cooperativity in drug-target interactions. The discussion would be strengthened if the authors described how these specific conclusions contribute to our ability to predict and combat mechanisms of resistance. These comments are needed because the presented model is the most successful at predicting in vivo drug activity when known resistance genes are deleted (Figure 4).

Some minor points:

10. Related to comment 6. On page 8, line 206-207, the authors compare the K_m values of PG precursor enzyme MraY from Gram-positives and -negatives to explain Figure 1e. Is this the rate limiting step in PG synthesis? Were other enzyme rates compared?

11. In the discussion, the authors suggest that MurJ, a Lipid II flippase, may accept UPP (lipid carrier) as a substrate since MurJ recognizes substrate through the pyrophosphate and sugar moieties (page 18, lines 457-459). Since possible contacts with sugars found in Lipid II (see also Kuk et al., *Nat. Struct. Molec Biol.* 2017, 24, 171) are absent in UPP, it is not apparent that MurJ would also recognize UPP.

12. The wording in the caption for Figure 2d is confusing, please clarify.

Reviewer #2:

Remarks to the Author:

This manuscript describes a theoretical model of the lipid II cycle of bacterial cell wall biosynthesis. By inputting literature parameters into their model, the authors show that the distribution of intermediates in the cycle is highly asymmetric, suggesting a possible explanation for why some cell-wall targeting antibiotics have in vivo MIC values close to their in vitro binding affinities while others do not. The model shows various interesting features such as an important role for the co-operativity of antibiotic binding, which are in agreement with literature observations. The model is used to predict MIC values for 5 different antibiotics in Gram positive organisms, with the results being in quite good agreement with literature.

The concept put forward here is interesting and the manuscript is very thorough and careful, although the narrative is sometimes hard to follow.

It is a pity that the experimental data that is used for comparison essentially amounts to only 5 MIC values (Fig 4): the paper would be much stronger with additional experimental data. Presumably this is what the literature provides, however, and it would be unreasonable to demand that every modelling paper be combined with in-house experiments.

Specific comments:

- the second paragraph of the introduction seems to imply that the glycopeptides studied here represent the entirety of cell wall targeting antibiotics. This is misleading, since there are many other types of cell wall targeting antibiotics including beta-lactams that act quite differently. The field should be described more comprehensively here.
- likewise an overview of the whole cell wall synthesis pathway, not just the lipid II cycle, should be provided (eg as a diagram).
- some of the claims made are over-blown. "No theoretical description of [cell wall synthesis] exists to date", "incorporating all existing evidence in the literature etc", "integrates all existing biochemical knowledge" etc. How can the authors be sure of these statements?
- The first paragraph of the results part actually belongs in the introduction.
- The introduction should make clear that the 5 chosen antibiotics all bind to lipid II intermediates and explain which intermediate each binds to.
- The model is first parameterized for E. coli then transformed to apply to Gram positives. It is not explained clearly why. Is this because suitable parameter values do not exist for Gram positives, but these antibiotics are not functional in Gram negatives?
- Lines 132-134: it is stated that because each carrier undergoes 24 transport cycles per cell division, it must be the pacemaker of cell wall synthesis. The logic is not clear here.
- The paper relies heavily on parameter values taken from the literature. It should therefore discuss how accurately these are known, how they have been measured, and what the sensitivity of the model predictions to errors in the parameter values is.
- on p.9 the claim is made that the model's predictions are "parameter-free". This is not the case - the predictions rely heavily on parameter values taken from the literature!

- in comparing model predictions with experimental literature data (for MICs) many imprecise statements are made (eg "in excellent agreement"). It would be much better to quantify the agreement in a well-defined way.
- The "first principles prediction" claims to discuss all 5 antibiotics but actually only 2 are discussed here and the others relegated to the SI. But surely these are key results?
- For the analytical calculations, a reduced model of the cycle is used in which all intermediates except 1 are lumped together. But is this really a good model? Surely some aspects will be different compared to the real cycle, for example in the real cycle there is presumably a time-delay created by the cycle which would not be present in the lumped together model.
- lines 347-348 references are needed
- Perhaps the most important figure, Fig 4, is not discussed at all in the Results and only features in the Discussion.
- the final sentence of the first paragraph of the Discussion seems highly speculative and also the logic of it is hard to follow. Why would multiple penicillin binding proteins prevent blocking by lipid II-binding antibiotics?

Reviewer #3:

Remarks to the Author:

Piepenbreier et al present a theoretical framework to better understand the reason behind discrepancies between antibiotic resistance patterns observed in vivo with those measured in vitro ("the efficacy gap") with a focus on antibiotics that target cell-wall biosynthesis. To this end, they first construct a kinetic model of the cell-wall biosynthesis for *E. coli* as a representative gram-negative bacterium and then adjust this model for gram-positive bacteria. This kinetic model is next coupled with a simplified model of antibiotic action, where they model the binding of an antibiotic with externally exposed intermediates of the cell-wall biosynthesis pathway. The authors then use this integrated model to assess the impact of five different antibiotics on the gram-positive model organism *Bacillus subtilis*. A key result from this theoretical analysis is that the asymmetric distribution of externally accessible metabolites in the cell-wall biosynthesis pathway can explain the large in vivo efficacy gap for these antibiotics. The second important result of this study is that the multimeric complex formation of antibiotics with their target (antibiotic-target cooperatively) can enhance the in vivo efficiency of the antibiotic by more efficiently sequestering the target.

The main contribution of this paper is to provide a mechanistic explanation for the observed large efficacy gap for antibiotics targeting cell-wall biosynthesis. Given the ever-increasing importance of addressing antibiotic resistance, theoretical studies like this work are valuable as they can shed light onto aspects of antibiotic actions that are not yet experientially accessible. Therefore, I believe that this work would be of interest to the broad readership of *Nature Communications*, however, I encourage the authors to address the following concerns to both clarify their contribution and to improve the quality of their manuscript:

MAJOR CONCERNS

1) The authors claim to have developed the “first” quantitative kinetic model for lipid II cycle and cell wall biosynthesis (lines 68-70 and line 116). This model was constructed for *E. coli* (a gram-negative bacteria) and then adjusted for gram-positive bacteria. There are, however, large-scale kinetic models of *E. coli* developed recently, which do account for lipid biosynthesis (see PMID: 27996047).

Therefore, the main contribution of the work presented by the authors seems to be integration of such a kinetic model of cell wall biosynthesis with antibiotic actions and not the development of the kinetic model itself. This has to be clarified in the manuscript.

2) The kinetic model of the cell-wall biosynthesis for *E. coli* has been constructed based on a Michaelis-Menten kinetics where the constants of this equation (K_m and V_{max}) have been obtained from literature or by solving a constrained optimization problem. It is, however, largely known that the in vitro measured kinetic parameters may be quite different from those in vivo and they may even change from one condition to another (e.g., for different growth and environmental conditions and for different genetic backgrounds) (see PMID: 21920040 & PMID: 23450699). Furthermore, for some kinetic parameters such as K_{cat} , there is a large variability in its reported values by different groups. While the authors have performed a robustness analysis for their model with respect to the values of parameters estimated by constrained optimization, their analysis does not include the measured parameter values extracted from literature. I encourage the authors to perform this analysis with respect to all kinetic parameter values (measured values reported in literature + estimated by using constrained optimization) to provide a more complete assessment of the model robustness, or to include a discussion about this in the text at minimum.

3) Given that the development of the quantitative integrated model of the cell-wall biosynthesis and antibiotic action is the core component of this study, I feel that more details about this model should be presented in the main text (while unnecessary details could be still kept in supplementary text).

MINOR CONCERNS

4) Parameters MIC, IC₅₀ and KD (dissociation constant), binding affinity (lines 56-57), binding constant (lines 264 and line 357) all seem to have been used in this manuscript interchangeably to refer to a minimal effective concentration of the antibiotic. While I understand that some of these parameters (like MIC and IC₅₀) have been commonly used in antibiotics-related literature, it will be helpful for broader audience to minimize reference to multiple terminologies, or otherwise to clarify which model parameters you would like to compare with reported values in literature to determine the efficacy gap or the accuracy of your model. In addition, note that binding affinity is not exactly the same as KD. In fact, it has an inverse relationship with KD.

5) For $A + T \rightleftharpoons AT$, KD is equal to concentration of A (antibiotic) at which half of T (in this case a lipid cycle intermediate) has formed a complex with A (in the form AT). By expecting KD to be equal to IC₅₀ (lines 250 – 252), do you imply that the PG synthesis drops down to 50% of its maximum if the half of a lipid cycle intermediate is sequestered in the form AT? Is this true for all lipid cycle intermediates? Why? Furthermore, the statement that the predicted IC₅₀ by the model is very similar to the in vitro KD (lines 259 – 260) contradicts with the statement in lines 250 – 251. Overall, it is unclear to me what the message of this paragraph (lines 244-260) is. I encourage the author to consider revisiting this paragraph.

6) I recommend the authors to just mention key elements of the lipid II cycle in the main text and to move the rest of the details presented in lines 90-106 to the caption of the corresponding figure as I do not think the details of this pathway are of interest to broader audience.

7) While the statement that the substrate affinity ($1/K_m$) has an inverse relationship with K_{cat} is true (lines 213-215), $K_m \sim K_1/K_{cat}$ is incorrect and must be inverted. The Michaelis-Menten constant is defined as $K_m = (K_{-1} + K_{cat})/K_1$. If $K_{cat} \gg K_{-1}$, then we have $K_m = K_{cat}/K_1$ implying that $1/K_m$ (substrate affinity) has an inverse relationship with K_{cat} .

8) Please cite PMID: 18577389, which is a relevant work on the computational study of antibiotics targeting cell wall biosynthesis.

9) I suggest to avoid using "dissociation constant" for K_{diss} as K_D is also referred to as dissociation constant. Furthermore, it would be wise to avoid using K_{ass} as "ass" is not the best abbreviation for association! For example, the authors can use K_{fwd} and K_{rev} instead of K_{ass} and K_{diss} .

10) Please provide relevant references for the statements= in lines 348-350.

11) Please avoid using abbreviations like "approx.". Use "approximately" instead.

Detailed response to reviewers

Reviewer #1 (Remarks to the Author):

Piepenbreier et al. present a combination of theoretical and experimental data to explain observed discrepancies between the *in vitro* and *in vivo* efficacy of several antibiotics. The authors focus on clinically-important antibiotics that bind to precursors of the essential cell wall that surrounds nearly all bacteria and is composed of peptidoglycan (PG). The authors use published data on reaction rates and precursor concentrations to establish their model in Gram-negative (*E. coli*) and then Gram-positive (*B. subtilis* and pathogens like *S. aureus*) bacteria. The authors present experimental data demonstrating that an antibiotic that inhibits early stages in precursor synthesis (lipid carrier) shows better agreement between *in vitro* and *in vivo* efficacy than an antibiotic that targets later stages (the PG monomer Lipid II) due to the “circular flux” nature of PG biosynthesis. Lastly, they illustrate that incorporating cooperativity factors for several drug-target interactions into mathematical models leads to successful predictions of *in vivo* drug activity, rather than simply using hyperbolic binding models for all antibiotics.

Overall, this work is clearly written, the models are explained with sufficient detail for biochemists/microbiologists, and the conclusions drawn from these data are significant, as they explain discrepancies between *in vitro* data and *in vivo* observations that are the subject of much debate for this class of antibiotics. However, some parameters of the model (and some of the discussion text) overlook key biological processes involved in PG synthesis and inhibition across different organisms. The manuscript would be significantly strengthened if the following major points were addressed:

1. In the introduction, the authors state “By incorporating all existing experimental evidence in the literature, our theory accounts for all biochemical knowledge of this essential pathway and reconciles it with the *in vivo* inhibition patterns under antibiotic treatment.” (pg 3, lines 73-75). This statement overreaches the scope of this work, and should be adjusted. The authors only test antibiotics that bind peptidoglycan precursors, and not antibiotics that target peptidoglycan biosynthetic enzymes (such as the PBPs). For example, the beta-lactams, which bind the PBPs, are a very important class of cell wall antibiotics but are not discussed in this paper. This should be clarified in the introduction. Other comments below only re-iterate this point about the true generality of this work.

We thank the reviewer for this comment and specified statements about the generality of our work accordingly. In the mentioned section and in further paragraphs of the manuscript, we pointed out more clearly that we focused on antibiotics binding to lipid II cycle intermediates (“substrate-sequestering antibiotics”) exclusively. The reason why we did not focus on enzyme-inhibiting antibiotics is that most enzymes in the cycle, and in particular the PBPs, operate redundantly. When blocking PBPs by beta-lactams, for instance, reliable predictions can only be made if one knows the precise protein abundances, kinetic rates, K_m values and beta-lactam binding affinities for *all* redundant PBPs in the cell, because they jointly dictate how a given drug concentration translates into a reduction of the *effective* transglycosylation rate, which is the relevant parameter to our model. Given that this picture is still incomplete on the experimental end, we feel that it is too early to quantitatively attempt predictions of MICs for beta-lactams. We further dwell on this in our response to the next point.

2. In line with the comment above, it would be helpful if the authors explained if their model could be used to describe the behavior of drugs such as the beta-lactams (covalent inhibitors) and moenomycins (non-covalent inhibitors), which target the PBPs (PG biosynthetic enzymes). For example, does the effect of PBP inhibition on PG monomer (Lipid II substrate) levels have a similar effect to drugs that bind Lipid II (like nisin)?

We totally agree that beta-lactams are an important class of antibiotics, that can, in principle, be studied within our model. However, as mentioned above, the inhibitory effects of beta-lactams on the lipid II cycle cannot be trivially predicted, because not all characteristics of the redundant PBPs are known.

Nevertheless, we started to study the inhibitory effect of other enzyme-binding antibiotics within our model by focussing on drugs that target reactions catalysed by only a single enzyme. For example, when analysing the effect of tunicamycin (inhibiting MraY and thereby the formation of lipid I), our model predicts that the IC_{50} is almost identical to the *in vitro* K_D – in line with experimental MIC measurements in *B. subtilis*. Similarly, if we treat all PBPs as one “effective enzyme” the model does not predict a discrepancy between the effective K_D of the beta-lactams and the predicted IC_{50} – although experimental verification of this prediction is difficult for the above reason. These results suggest that the predicted buffering effect is a phenomenon exclusively relevant for antibiotics binding to lipid II cycle intermediates and not found for enzyme-inhibiting antibiotics.

Furthermore, we also started to study the effects of drug-drug combinations within the model. Here, we analysed (i) antibiotic combinations composed of two antibiotics featuring the same inhibition mechanism and (ii) combinations of enzyme- and substrate-inhibiting antibiotics. Preliminary results show that a combination of an enzyme-inhibiting

antibiotic and an antibiotic that binds the substrate of the respective enzymatic reaction leads to nonlinear effects on the lipid II cycle, indicating synergistic interactions between the drugs.

Given that these predictions clearly require downstream experimental verification, which is far beyond the scope of the current manuscript, we would like to defer a detailed analysis and discussion about enzyme-inhibiting antibiotics (and their interaction with substrate-binding antibiotics) to a follow-up manuscript. Thus, we hope the reviewer will agree that focussing the present manuscript on the class of substrate-binding antibiotics, their important feature of the *in vivo* efficacy gap and the effect of cooperative binding are sufficiently rich in novel insights to warrant publication as a closed story.

3. The authors compare the dimensions of the peptidoglycan layer of Gram-negative and -positive bacteria in order to create a mathematical model for PG synthesis for Gram-positives based on the model built for Gram-negatives. The authors state that Gram-positives must require an increased rate of PG synthesis because their cell walls are thicker than that of Gram-negatives but the cells are similarly sized (page 8, lines 191-197). However, a potential role for PG hydrolases in differences between Gram-positive and -negative cell wall thickness is never raised. The introduction and discussion sections, as well as the derivation of the model, would be strengthened if the possible effect of PG degradation enzymes were acknowledged.

In order to estimate the amount of PG required to build the walls of Gram-positive and Gram-negative organisms, we actually considered the PG turnover rates in the original model. Specifically, we adapted the flux of PG precursors that needs to be shuttled across the membrane within one generation time (j_{PG}) to match the PG required to build a new sacculus *plus* the portion of PG that is degraded by hydrolases. The calculation is detailed in the Supplementary Text in the first paragraph of the section *Quantitative considerations on peptidoglycan synthesis in E. coli* and in Supplementary Table 3a.

However, we totally agree that we missed to state this in the main manuscript. To improve presentation of this aspect, we now specify the description of the cell wall turnover in the Supplementary Text and highlight the role of hydrolases in the model description in the main text (lines 130/131) and in the discussion (p. 20, lines 510-513).

4. A defining difference between Gram-negatives and -positives is the presence of an outer membrane in the former. The authors note that one possible (and popular) explanation for discrepancies between *in vitro* and *in vivo* efficacies is “sequestration” (line 61, page 3) or a change in effective concentration *in vivo*. To broaden the scope of this work, the authors could discuss if their model predicts changes in drug activity measured for Gram-negatives with and without permeabilized outer membranes to explore the validity of the “sequestration” effect. An outer membrane deficient *E. coli* strain (see Eggert et al., *Science*, 2001, 294, 361) may be worthwhile to test experimentally following Figure 1.

Yes – in Gram-negatives the outer membrane poses a permeability barrier, which is the reason why most of the antibiotics targeting the intermediates of lipid II cycle (often cationic antimicrobial peptides) are ineffective against Gram-negatives (see e.g. PMID 26523408). We agree that testing our model predictions in an outer membrane-deficient *E. coli* strain would be insightful in the future, but we hope the reviewer will agree that such experiments are beyond the scope of this theoretical study. Also, we feel that discussing the validity of the sequestration effect in Gram-negatives +/- outer membrane may lead away from the core of this work and may rather confuse the reader, which is why we decided to not further dwell on this point in the manuscript.

5. In line with the comment above, why are only antibiotics typically used to treat Gram-positives tested in this paper? This should be explained in the introduction and/or discussion when relevant (especially when making broad conclusions from this work).

To our knowledge, there are no antibiotics other than beta-lactams that interfere with the lipid II cycle in Gram-negatives. We now state this in the text (lines 119-121) and explain that modelling beta-lactam activity will require incorporation of the biochemical characteristics of *all* redundant PBPs (lines 505-510).

6. The authors state that they “integrate all known biochemical properties” (page 7, line 184) to derive their model; however, this statement is not convincing based on the information provided. For example, in Supplementary Table 2, footnote b states “kcat for the bifunctional transglycosylase PBP1b (only available kcat)”. In the text, the authors note that characterization of Gram-positive PBPs is “sparse” (page 7, lines 186-187). There have been many kinetic studies done on PBPs from various organisms. For a quick review, please see Sauvage, E. et al. (*FEMS Microbiol Rev.* 2008, 32, 234) Table 1, which contains kinetic parameters for six PBPs (from a mixture of Gram-negative and Gram-positive organisms). Most significantly, PBP1a is listed, which is considered as a major bifunctional PG synthetic enzyme (along with PBP1b) in *E. coli*, and is important to consider in any unified PG model. If the authors excluded this data for a particular reason, it should be explained.

We thank the reviewer for this important comment and the suggested reference, and took the additional data measured for *E. coli* into account. Since the K_M values measured for PBP1a and PBP1b range around the same value

(2 μ M), there was no need to change this parameter value in the model, which was already set to 2 μ M before. Nevertheless, we show in our newly performed parameter sensitivity analysis (as suggested by Reviewers 2 and 3) that variation of the K_M value by a factor of two does not significantly affect the model predictions (new Fig. S5). Strikingly, we noticed that the k_{cat} values for the same PBP (PBP1b) sometimes differed orders of magnitude between measurements reported in different studies. Apart from the fact k_{cat} values were not available for all enzymes of the lipid II cycle, this was one reason not to include the k_{cat} values as direct inputs for the model but rather used them to validate the parameter fitting results. Thus, this parameter does not have any impact on the model prediction and we decided to show the results for one of the k_{cat} values exemplarily. However, we changed the statement in Supplementary Table 2 mentioned by the Reviewer.

Finally, we decided to not include the biochemical data on PBPs from the other organisms mentioned in Sauvage et al. The reason for this is that due to the high redundancy between PBPs, it is often unclear which of the PBPs plays the most dominant role in a given organism and under and given growth condition. Thus, given that even homologous PBPs do not necessarily have the same importance in different organisms, we felt it would be highly speculative to compare the biochemical properties of only a few (potentially non-comparable) PBPs between organisms.

7. In Figure 2b and Supplementary Figure 3, how were Lipid II cycle intermediate concentrations measured with changing antibiotic concentration? Supplementary Table 1 references past reports for levels of intermediates in *E. coli*; were the same techniques employed?

We are sorry that there was a misconception about these data. The concentrations of lipid II cycle intermediates under different antibiotic concentrations (in Figures 2b and S3) are *predictions* of our theoretical model, which we now state explicitly in the corresponding figure legends.

8. Please include a procedure for determining MIC values in the supplemental (or list references in a separate section of the supplemental).

Since all MIC data were taken from literature, we did not experimentally determine MICs ourselves. All references for the MIC values are listed in the Supplementary Table 4b. In addition, we specified the approach that was used to determine the MIC in the diverse studies, as suggested, by including additional information about the method and the media that were used (Supplementary Table 4b).

9. The authors suggest at the end of the discussion that this work may help combat resistance. This work clearly describes the ability of cells to compensate for inhibition of substrates by adjusting pathway flux and takes into account cooperativity in drug-target interactions. The discussion would be strengthened if the authors described how these specific conclusions contribute to our ability to predict and combat mechanisms of resistance. These comments are needed because the presented model is the most successful at predicting in vivo drug activity when known resistance genes are deleted (Figure 4).

We totally agree and revised the last paragraph of the discussion accordingly (lines 498-528):

“... Hence, we believe that the presented model serves as an excellent starting point to develop a “whole cell model” of antibiotic action. One important aspect will be the development of theoretical models describing the regulation and action of known resistance mechanisms (which are deleted from in the strains considered in this work), to provide a systems-level description of antibiotic action wildtype cells. First steps in developing such models have been made, e.g., for the bacitracin resistance determinant BceAB in B. subtilis (80) and for beta-lactamases in S. aureus (81).”

Some minor points:

10. Related to comment 6. On page 8, line 206-207, the authors compare the K_M values of PG precursor enzyme MraY from Gram-positives and -negatives to explain Figure 1e. Is this the rate limiting step in PG synthesis? Were other enzyme rates compared?

The K_M of MraY was the only value measured in *B. subtilis* we found in the literature. As mentioned in the response to Comment 6, we did not include the measurements for the K_M values of PBPs for other Gram-positives into our comparison for the given reasons.

11. In the discussion, the authors suggest that MurJ, a Lipid II flippase, may accept UPP (lipid carrier) as a substrate since MurJ recognizes substrate through the pyrophosphate and sugar moieties (page 18, lines 457-459). Since possible contacts with sugars found in Lipid II (see also Kuk et al., Nat. Struct. Molec Biol. 2017, 24, 171) are absent in UPP, it is not apparent that MurJ would also recognize UPP.

Although Kuk et al. (2017) describe the contact of the GlcNAc and MurNAc moieties of lipid II with MurJ, a more recent study (Bolla, J.R. et al. Nat Chem (2018), cited in our discussion) additionally identifies the pyrophosphate moiety as part of the interaction interface between lipid II and MurJ. We therefore feel that it is plausible that MurJ may have a somewhat lower affinity also for UPP, by recognizing its pyrophosphate headgroup. Given that in the cell UPP

is ~100-fold more abundant than lipid II, we also think that this high concentration may compensate for a potentially lower binding affinity to MurJ, and could be sufficient to warrant UPP binding to and subsequent transmembrane transport by MurJ.

To better motivate these thoughts in the discussion, we re-wrote the sentence as follows (lines 469-473):
“A recent study in *E. coli* revealed, however, that the lipid II flippase MurJ recognizes its substrate via the pyrophosphate, GlcNAc and MurNAc moieties (77), raising the possibility that MurJ could also recognize UPP (via its pyrophosphate headgroup) as a secondary substrate, albeit with lower affinity, and license it for trans-bilayer transport.”

Reviewer #2 (Remarks to the Author):

This manuscript describes a theoretical model of the lipid II cycle of bacterial cell wall biosynthesis. By inputting literature parameters into their model, the authors show that the distribution of intermediates in the cycle is highly asymmetric, suggesting a possible explanation for why some cell-wall targeting antibiotics have in vivo MIC values close to their in vitro binding affinities while others do not. The model shows various interesting features such as an important role for the co-operativity of antibiotic binding, which are in agreement with literature observations. The model is used to predict MIC values for 5 different antibiotics in Gram positive organisms, with the results being in quite good agreement with literature.

The concept put forward here is interesting and the manuscript is very thorough and careful, although the narrative is sometimes hard to follow.

It is a pity that the experimental data that is used for comparison essentially amounts to only 5 MIC values (Fig 4): the paper would be much stronger with additional experimental data. Presumably this is what the literature provides, however, and it would be unreasonable to demand that every modelling paper be combined with in-house experiments.

Specific comments:

- the second paragraph of the introduction seems to imply that the glycopeptides studied here represent the entirety of cell wall targeting antibiotics. This is misleading, since there are many other types of cell wall targeting antibiotics including beta-lactams that act quite differently. The field should be described more comprehensively here.

We thank the reviewer for this important point and revised the second paragraph of the introduction accordingly (lines 80-84). As explained in the response to Comment 2 of Reviewer 1 in detail, we think it is important to focus in this study of the class of substrate-inhibiting antibiotics, since they feature significant *in vivo* efficacy gaps. In the near future, we aim to broaden our analyses and include enzyme-inhibiting antibiotics and study their interaction by substrate-inhibition antibiotics.

- likewise an overview of the whole cell wall synthesis pathway, not just the lipid II cycle, should be provided (eg as a diagram).

We carefully thought about this suggestion, but came to the conclusion that we would like to limit the graphical presentation to the lipid II cycle, as well as a schematic representation of the cytoplasmic PG precursor synthesis, as shown in the original Fig. 1. Although we understand that depicting other pathways, such as cytoplasmic bactoprenol, WTA or LPS synthesis, or PG recycling pathways could help to draw a comprehensive picture of cell envelope synthesis, we feel that including this level of detail will be confusing to the reader, since it is not the focus of the present work. Therefore, we hope that the reviewer will consent to our idea that there are pros and cons to both options, and hope that he/she will agree that the option we chose will be the more focussed option.

- some of the claims made are over-blown. "No theoretical description of [cell wall synthesis] exists to date", "incorporating all existing evidence in the literature etc", "integrates all existing biochemical knowledge" etc. How can the authors be sure of these statements?

We agree and since this has also been noted by the other reviewer's, we toned down our claims regarding novelty and exclusiveness throughout the manuscript, now providing a more neutral wording in our presentation (lines 88-89, 159).

- The first paragraph of the results part actually belongs in the introduction.

We moved the basic description of the lipid II cycle into the introduction section, as suggested (lines 68-78). In addition, some of the details, e.g. on the extracytoplasmic activity of UPP phosphatases, are shifted to the caption of Figure 1, as suggested by reviewer 3.

- The introduction should make clear that the 5 chosen antibiotics all bind to lipid II intermediates and explain which intermediate each binds to.

In line with the first comment of the reviewer, we revised the second and third section in the introduction that both describe the antibiotics we focus on in our study (lines 80-84 and 93-96).

- The model is first parameterized for *E. coli* then transformed to apply to Gram positives. It is not explained clearly why. Is this because suitable parameter values do not exist for Gram positives, but these antibiotics are not functional in Gram negatives?

The reviewer is completely right. We re-worded the paragraph for better clarity (lines 107-123):

*“The bacterial cell wall consists of an alternating polymer of N-acetylglucosamine (GlcNAc) and N-acetylmuramic acid (MurNAc), cross-linked by a MurNAc-attached pentapeptide (Fig. 1a) (27, 28). Even though Gram-negative and -positive bacteria greatly vary in cell wall thickness and some organisms show specific modifications in peptidoglycan composition (e.g. variations in the GlcNAc-MurNAc-pentapeptide known for Staphylococci) or cross-linking properties (e.g. in Corynebacteria) (36), the central lipid II cycle of cell wall biosynthesis is highly conserved throughout the bacterial world (Fig. 1a). Accordingly, it seems plausible that the basic working principles of the lipid II cycle are similar between Gram-negative and Gram-positive bacteria. Most biochemical work on the enzymes and intermediates of the lipid II cycle, however, was focussed on the Gram-negative model organism *E. coli*. Therefore, in the following we will first perform some general considerations on the kinetics of cell wall synthesis in *E. coli*, which will lead us to a first quantitative model for this essential process in Gram-negatives. Given that most antibiotics targeting the intermediates of the lipid II cycle are ineffective against Gram-negatives (due to the permeability barrier posed by the outer membrane), in a second stage we will adapt the model to Gram-positive-specific cell wall synthesis, which will allow us to make testable predictions for cell wall antibiotic action in *B. subtilis* and other Gram-positive bacteria.”*

- Lines 132-134: it is stated that because each carrier undergoes 24 transport cycles per cell division, it must be the pacemaker of cell wall synthesis. The logic is not clear here.

We reworded the sentence for better clarity, as follows:

“Thus, each carrier undergoes an average of ~24 transport cycles before it gets diluted due to cell growth. This suggests that instead of synthesizing lipid carriers for one-time “use it and lose it” transport, lipid carrier recycling is the pacemaker of PG monomer transport across the membrane. Under these conditions the lipid II cycle can be approximated as a closed-loop system, in which the pool levels of lipid II cycle intermediates quickly equilibrate, leading to cyclic flux-balance between all of the states, i.e. $j_1 = j_2 = \dots = j_6$ (Fig. 1b, blue arrows; Supplementary Text).”

- The paper relies heavily on parameter values taken from the literature. It should therefore discuss how accurately these are known, how they have been measured, and what the sensitivity of the model predictions to errors in the parameter values is.

We totally agree that it is highly important to study the sensitivity of the model predictions not only towards variations in the fit parameters (as done in detail in the original submission; results shown in Fig. S4) but also to possible variations in the parameter extracted from literature. To complete the analysis on this end, we performed a sensitivity analysis of the *model* predictions towards variations in all K_M values and the measurements of pool levels of the various lipid II cycle intermediates, as taken from the literature (new Fig. S5). The sensitivity analyses revealed that model predictions vary only slightly under variations in all of these parameters, i.e. 100% changes in the model parameters generally lead to <100% changes in the corresponding IC50 value predictions for the different antibiotics. This shows that as long as experimental measurements are not off by orders of magnitude, our predicted IC50 values will still be in the same ballpark as the experimental MICs.

- on p.9 the claim is made that the model's predictions are "parameter-free". This is not the case - the predictions rely heavily on parameter values taken from the literature!

We agree that the statement was misleading and corrected accordingly:

“This allowed us to integrate these binding reactions into our quantitative model for the lipid II cycle (see Supplementary Text for the description of the modelling of antibiotic-target interactions) – thereby creating a tool to generate predictions of cell wall antibiotic action without invoking further fit parameters.”

- in comparing model predictions with experimental literature data (for MICs) many imprecise statements are made (eg "in excellent agreement"). It would be much better to quantify the agreement in a well-defined way.

We thank the reviewer for this comment and quantified the respective statements accordingly (lines 296, 303, and 388).

- The "first principles prediction" claims to discuss all 5 antibiotics but actually only 2 are discussed here and the others relegated to the SI. But surely these are key results?

Yes - the section *Predicting cell wall antibiotic action from first principles* only discusses two of the five antibiotics (the non-cooperative binders bacitracin and nisin), but the other three (vancomycin, ramoplanin and friulimicin) are discussed in detail in the section *Cooperative drug-target interaction drastically boosts antibiotic efficacy in vivo*. Thus, although we did not show the plots of antibiotic concentration vs. PG synthesis rate (Fig. 2a, c) for all five antibiotics, we discussed them in detail in the results section and show the final IC_{50} predictions in Figure 4. To make our presentation more transparent we now explain our rationale in the introducing paragraph to the section "Predicting cell wall antibiotic action from first principles" (lines 249-251):

"In the following we will first focus on two antibiotics that bind their target non-cooperatively and in a later section consider the effect of cooperative binding for the remaining three antibiotics."

In addition, we now provide more frequent cross-links to Fig. 4, which summarizes the key results for the remaining three antibiotics.

- For the analytical calculations, a reduced model of the cycle is used in which all intermediates except 1 are lumped together. But is this really a good model? Surely some aspects will be different compared to the real cycle, for example in the real cycle there is presumably a time-delay created by the cycle which would not be present in the lumped together model.

The reviewer is absolutely right that there are some quantitative differences to be expected between the full and the reduced model of the lipid II cycle. Apparently, the reviewer may have missed that we already discussed the similarities and differences between the full model and the reduced model in detail in the Supplementary Text (last paragraph of the section *Reduced model of the lipid II cycle*). Here we carefully studied the impact of the simplifying assumption of the reduced model on the model predictions and showed that the model predictions of the reduced model closely resemble the model predictions of the full model (Fig. S3g). Please note that we did not use the reduced model to predict the buffering effect itself, but rather to find an analytical expression that help to rationalize this effect. It is clear that this is not 100% precise (due to some simplifying assumptions of the reduced model) but it captures the essence of the mechanism and is exclusively used to illustrate the buffering effect. Therefore, we believe that this mathematical expression is quite insightful and the reduction of the model is necessary to find an analytical solution.

- lines 347-348 references are needed

We added the missing references.

- Perhaps the most important figure, Fig 4, is not discussed at all in the Results and only features in the Discussion.

We revised this and referenced Figure 4 whenever discussing the results of *B. subtilis* in the Results section (p. 10, line 255; p. 10, line 270; p.11, line 292; p. 11, line 295; p. 11, line 301; p.12, line 305; p.14, line 382; p. 15, line 388; p.16, line 421) and exclusively discussed the results of other pathogen Gram-positives displayed in Figure 4 in the Discussion (p. 18, lines 452-461).

- the final sentence of the first paragraph of the Discussion seems highly speculative and also the logic of it is hard to follow. Why would multiple penicillin binding proteins prevent blocking by lipid II-binding antibiotics?

The idea was that high PBP activity will increase the rate of the transglycosylation reaction from lipid II to UPP (+wall-linked PG monomers) and thereby decrease the lipid II pool. However, we agree that this does not require multiple (different) PBPs, but rather sufficiently high enzyme activities and copy numbers. To make this point more transparent and avoid confusion with the redundancy of PBPs, we simplified our statement as follows:

"In this light it seems plausible that bacteria may have evolved to minimize the abundance of externally exposed lipid II molecules, e.g. by speeding up the rate of PG monomer insertion into the cell wall, in order to evade blocking by lipid II-binding antibiotics, which are ubiquitously produced by competing species, such as Lactobacillus lactis (nisin), Amycolatopsis orientalis (vancomycin) or Actinomycetes species (ramoplanin) (71, 72 73, 74)."

Reviewer #3 (Remarks to the Author):

Piepenbreier et al present a theoretical framework to better understand the reason behind discrepancies between antibiotic resistance patterns observed in vivo with those measured in vitro (“the efficacy gap”) with a focus on antibiotics that target cell-wall biosynthesis. To this end, they first construct a kinetic model of the cell-wall biosynthesis for *E. coli* as a representative gram-negative bacterium and then adjust this model for gram-positive bacteria. This kinetic model is next coupled with a simplified model of antibiotic action, where they model the binding of an antibiotic with externally exposed intermediates of the cell-wall biosynthesis pathway. The authors then use this integrated model to assess the impact of five different antibiotics on the gram-positive model organism *Bacillus subtilis*. A key result from this theoretical analysis is that the asymmetric distribution of externally accessible metabolites in the cell-wall biosynthesis pathway can explain the large in vivo efficacy gap for these antibiotics. The second important result of this study is that the multimeric complex formation of antibiotics with their target (antibiotic-target cooperatively) can enhance the in vivo efficiency of the antibiotic by more efficiently sequestering the target.

The main contribution of this paper is to provide a mechanistic explanation for the observed large efficacy gap for antibiotics targeting cell-wall biosynthesis. Given the ever-increasing importance of addressing antibiotic resistance, theoretical studies like this work are valuable as they can shed light onto aspects of antibiotic actions that are not yet experimentally accessible. Therefore, I believe that this work would be of interest to the broad readership of *Nature Communications*, however, I encourage the authors to address the following concerns to both clarify their contribution and to improve the quality of their manuscript:

MAJOR CONCERNS

1) The authors claim to have developed the “first” quantitative kinetic model for lipid II cycle and cell wall biosynthesis (lines 68-70 and line 116). This model was constructed for *E. coli* (a gram-negative bacteria) and then adjusted for gram-positive bacteria. There are, however, large-scale kinetic models of *E. coli* developed recently, which do account for lipid biosynthesis (see PMID: 27996047). Therefore, the main contribution of the work presented by the authors seems to be integration of such a kinetic model of cell wall biosynthesis with antibiotic actions and not the development of the kinetic model itself. This has to be clarified in the manuscript.

The reviewer is right that many researchers have developed large-scale kinetic models for metabolic pathways before us. We also appreciate that there have been models describing fatty acid and short-chain lipid biosynthesis (as e.g. PMID: 27996047). However, neither this work nor any other work we are aware of touches the lipid II cycle of cell wall biosynthesis modelled here (in fact, peptidoglycan synthesis is not even mentioned in PMID: 27996047). Yet, given that we cannot exclude that we may have missed previous mathematical modelling studies of the lipid II cycle we toned down our claims regarding exclusiveness and novelty, as uniformly suggested by all reviewers.

2) The kinetic model of the cell-wall biosynthesis for *E. coli* has been constructed based on a Michaelis-Menten kinetics where the constants of this equation (K_m and V_{max}) have been obtained from literature or by solving a constrained optimization problem. It is, however, largely known that the in vitro measured kinetic parameters may be quite different from those in vivo and they may even change from one condition to another (e.g., for different growth and environmental conditions and for different genetic backgrounds) (see PMID: 21920040 & PMID: 23450699). Furthermore, for some kinetic parameters such as K_{cat} , there is a large variability in its reported values by different groups. While the authors have performed a robustness analysis for their model with respect to the values of parameters estimated by constrained optimization, their analysis does not include the measured parameter values extracted from literature. I encourage the authors to perform this analysis with respect to all kinetic parameter values (measured values reported in literature + estimated by using constrained optimization) to provide a more complete assessment of the model robustness, or to include a discussion about this in the text at minimum.

We thank the reviewer for this thoughtful comment, which was also mentioned by Reviewer 2. As explained above in response to Reviewer 2, we totally agree that it is highly important to study the sensitivity of the model predictions not only towards variations in the fit parameters but also to possible variations in the parameters extracted from literature, as suggested. To address this point in more depth, we additionally performed a sensitivity analysis of the model predictions towards all experimentally determined model parameters (as explained in more detail above and in the legend to Fig. S5) and summarized the results in an additional Supplementary Figure (Fig. S5).

3) Given that the development of the quantitative integrated model of the cell-wall biosynthesis and antibiotic action is the core component of this study, I feel that more details about this model should be presented in the main text (while unnecessary details could be still kept in supplementary text).

We thought intensively about this point and see that the theoretically inclined reader may want to dig into the equations while reading the text. However, due to space constraints it will not be possible to show the comprehensive

mathematical derivation in the main text, which would force the inclined reader to jump back and forth between main text and Supplementary Information. Also, we feel that the majority of the audience may be discouraged from excessive math in the main text, and hence think the solution we chose – having a self-contained Supplementary Text providing the full model derivation on the one hand and the main text discussing the main features and working principle of the model on the other hand – will be most suitable for the majority of readers. This is also supported by the comment of Reviewer 1, stating that the description of the mathematical aspects are sufficiently detailed for biochemists/microbiologists, who we suppose will represent the largest audience for this manuscript.

MINOR CONCERNS

4) Parameters MIC, IC₅₀ and KD (dissociation constant), binding affinity (lines 56-57), binding constant (lines 264 and line 357) all seem to have been used in this manuscript interchangeably to refer to a minimal effective concentration of the antibiotic. While I understand that some of these parameters (like MIC and IC₅₀) have been commonly used in antibiotics-related literature, it will be helpful for broader audience to minimize reference to multiple terminologies, or otherwise to clarify which model parameters you would like to compare with reported values in literature to determine the efficacy gap or the accuracy of your model. In addition, note that binding affinity is not exactly the same as KD. In fact, it has an inverse relationship with KD.

We revised the manuscript (p. 3, line 57; p. 10, line 269; p. 14, line 369; p. 14, line 373; p.15, line 408; p.18, line 458, p. 33, line 779; p.35, line 839) and the Supplementary Text as suggested and limit ourselves to the parameters MIC, IC₅₀ and K_D (= dissociation constant) and the more general descriptive term *binding affinity* to clarify our statements.

5) For $A + T \rightleftharpoons AT$, KD is equal to concentration of A (antibiotic) at which half of T (in this case a lipid cycle intermediate) has formed a complex with A (in the form AT). By expecting KD to be equal to IC₅₀ (lines 250 – 252), do you imply that the PG synthesis drops down to 50% of its maximum if the half of a lipid cycle intermediate is sequestered in the form AT? Is this true for all lipid cycle intermediates? Why? Furthermore, the statement that the predicted IC₅₀ by the model is very similar to the in vitro KD (lines 259 – 260) contradicts with the statement in lines 250 – 251. Overall, it is unclear to me what the message of this paragraph (lines 244-260) is. I encourage the author to consider revisiting this paragraph.

We understand the confusion of the reviewer, since our initial wording suggested that for the UPP-binding antibiotic bacitracin we expect K_D and IC₅₀ to be of similar order of magnitude, which is not evident at this stage in the text. To avoid this confusion we rewrote the sentence in lines 258-260, as follows:

“To understand why the predicted IC₅₀ almost coincides with the K_D value in the model, we analysed the relative abundances of lipid II cycle intermediates at different bacitracin concentrations (Fig. 2b).”

This also resolves the apparent conflict with our later statement, in which we conclude that the model only predicts a marginal *in vivo* efficacy gap for bacitracin (now in lines 267-268).

6) I recommend the authors to just mention key elements of the lipid II cycle in the main text and to move the rest of the details presented in lines 90-106 to the caption of the corresponding figure as I do not think the details of this pathway are of interest to broader audience.

We thank the reviewer for this suggestion and implemented accordingly. Now the basics of the lipid II cycle are introduced in the introduction section and the detailed description is presented in the caption to Fig. 1, as suggested.

7) While the statement that the substrate affinity (1/K_m) has an inverse relationship with K_{cat} is true (lines 213-215), $K_m \sim K_1/K_{cat}$ is incorrect and must be inversed. The Michaelis-Menten constant is defined as $K_m = (K_1 + K_{cat})/K_1$. If $K_{cat} \gg K_1$, then we have $K_m = K_{cat}/K_1$ implying that 1/K_m (substrate affinity) has an inverse relationship with K_{cat}.

The Reviewer is completely right here and we corrected this mistake. The statement, however, stays unaffected (line 221).

8) Please cite PMID: 18577389, which is a relevant work on the computational study of antibiotics targeting cell wall biosynthesis.

We thank the reviewer for this suggestion and agree that references to work on modelling of resistance mechanisms were missing. To address this we included the following sentences in the discussion, including the suggested reference (lines 514-518):

"Hence, we believe that the presented model serves as an excellent starting point to develop a "whole cell model" of antibiotic action. One important aspect will be the development of theoretical models describing the regulation and action of known resistance mechanisms (which are deleted in the strains considered in this work), to provide a systems-level description of antibiotic action in wildtype cells. First steps in developing such models have been made, e.g., for the bacitracin resistance determinant BceAB in B. subtilis (80) and for beta-lactamases in S. aureus (81)."

9) I suggest to avoid using "dissociation constant" for K_{diss} as KD is also referred to as dissociation constant. Furthermore, it would be wise to avoid using K_{ass} as "ass" is not the best abbreviation for association! For example, the authors can use K_{fwd} and K_{rev} instead of K_{ass} and K_{diss} .

We thank the reviewer for this suggestion. However, since k_{ass} and k_{diss} are frequently used abbreviations for the rates of association and dissociation in the modelling world (see below for a reference), we think it will be consistent to keep these abbreviations. Nevertheless, we completely agree that k_{diss} should not be termed 'dissociation constant' but rather 'dissociation rate' and made sure that this mistake does no longer occur in the manuscript and the Supplementary Text (*changes only necessary in Supplementary Text*).

Determination of Association Rate Constants by an Optical Biosensor Using Initial Rate Analysis

Paul R. Edwards* and Robin J. Leatherbarrow^{†,1}

**Affinity Sensors, Saxon Way, Bar Hill, Cambridge, CB3 8SL, United Kingdom; and [†]Biological Chemistry, Department of Chemistry, Imperial College of Science, Technology and Medicine, South Kensington, London, SW7 2AY, United Kingdom*

Received April 23, 1996

[redacted]

10) Please provide relevant references for the statements= in lines 348-350.

Done.

11) Please avoid using abbreviations like "approx.". Use "approximately" instead.

Revised (p.10, line 258)

Reviewers' Comments:

Reviewer #1:

Remarks to the Author:

Reviewer #1's response:

Thank you for your responses to my points, the revised paper is very clear. I am satisfied with the authors' responses overall, but I have the following suggestions:

In response to point 6:

The logic of studying substrate-targeting antibiotics in the revised introduction is clear. I suggest that instead of saying that the biochemical data is "sparse" in Gram-positives, that the authors explain to the reader that an understanding of individual PBP function has not been laid out (line 193).

In response to point 11:

This is a direct quote from the Bolla et al, Nat Chem (2018) paper that the authors cite:

"Investigating further the specificity of lipid II binding to MurJ, we recruited a second potential lipid substrate, undecaprenyl phosphate (C55-P), which is a precursor in the synthesis of lipid II5 . Under similar MS conditions, we did not observe binding of C55- P even at high concentrations (30 μ M) (Supplementary Fig. 3d). This result, together the antibiotic binding data above, confirm that while the pentapeptide and undecaprenyl chain are not critical for binding, the pyrophosphate, MurNac and GlcNac moieties of lipid II are important for recognition by MurJ"

C55-P = UP, since MurJ doesn't bind UP, it is unlikely that it binds UPP.

Even though MurJ binds the phosphate portion of Lipid II, binding to the sugars likely ensures specificity (and contributes to binding energy). We can't assume that all phosphates bind MurJ (then MurJ might also accept Lipid I as a substrate, which would be problematic).

I think lines 469-473 should be removed, they are not essential.

Reviewer #2:

Remarks to the Author:

The authors have fully addressed the comments that I made in my previous report and have made significant changes that have improved the manuscript.

While I do not have any further major concerns, on reading the revised manuscript I have a number of more minor comments that the authors may wish to consider.

- in the introduction, lines 41-43 "most of the predictive models..." - this statement seems to contradict the examples presented above since only 1/3 of these models actually did use the ribosome growth laws as stated here.

- the 5 antibiotics that are studied are briefly listed in the introduction, but it is only stated that they act against intermediates in the lipid II cycle, whereas in fact it is crucial to know that they act against different intermediates, and which ones. I feel that a little more information here, and also a reference

to the diagram of Figure 1a, would be useful.

- lines 168-9 "we fixed ... to the experimental values" more accurate here would be to say they were fixed to values obtained from the literature, with references.

- line 178 "all biochemical...constraints" - this seems exaggerated, I would remove the "all"

- line 180 "biased" seems a confusing way to put this - "asymmetric" might be clearer.

- lines 205-6 "concentration per surface area" should be surface density, or surface concentration.

- line 213 " we find that the" - this is confusing as it sounds as though the authors have measured it, whereas it is actually from the literature. Better would be "literature suggests that the"

- line 223 "taken this and all existing" is overblown and should be toned down

- line 244 should contain a reference to the diagram in Fig 1a.

- line 275 "massive" - this seems exaggerated. Massive compared to what?

- line 315-6 it would be useful to state in more detail why the authors "wanted to derive an intuitive mathematical formula" - what are the advantages of this?

- line 318 I appreciate the authors' point that the supplementary text contains a discussion of the level of realism of the analytical model, but I think this should also be explained in the main text, ie rather than just pointing to the supp text one could write "see the supp text for a discussion of the realism of this analytical model"

- the buffering effect which is discussed extensively in the latter part of the results is a key result, but it is not necessarily so clear what is meant by this effect from the text. The authors could consider adding a simple diagram to illustrate what is meant by the buffering,

- line 57 "is" -> "are"

- line 82 "if the involves" -> "of the involved"

- line 250 "two antibiotics" -> "the two antibiotics"

- line 407 "not always confer" -> "do not always confer"

Reviewer #3:

Remarks to the Author:

The authors have satisfactorily addressed all of my concerns

Response to Reviewers:

We thank all reviewers for their constructive comments and suggestions, which we implemented as detailed below in our point-by-point response.

Kind regards,
Georg Fritz (of behalf of all authors)

REVIEWERS' COMMENTS:

Reviewer #1 (Remarks to the Author):

Reviewer #1's response:

Thank you for your responses to my points, the revised paper is very clear. I am satisfied with the authors' responses overall, but I have the following suggestions:

In response to point 6:

The logic of studying substrate-targeting antibiotics in the revised introduction is clear. I suggest that instead of saying that the biochemical data is "sparse" in Gram-positives, that the authors explain to the reader that an understanding of individual PBP function has not been laid out (line 193).

We implemented wording, as suggested (lines 191-194): "Even though for Gram-positive bacteria a comprehensive biochemical understanding of the PG synthetic machinery, and in particular of the PBPs, has not been laid out, we next integrated all existing quantitative knowledge from diverse species to consolidate them in a modified mathematical model for the Gram-positive cell wall synthesis."

In response to point 11:

This is a direct quote from the Bolla et al, Nat Chem (2018) paper that the authors cite:

"Investigating further the specificity of lipid II binding to MurJ, we recruited a second potential lipid substrate, undecaprenyl phosphate (C55-P), which is a precursor in the synthesis of lipid II5. Under similar MS conditions, we did not observe binding of C55-P even at high concentrations (30 μ M) (Supplementary Fig. 3d). This result, together with the antibiotic binding data above, confirm that while the pentapeptide and undecaprenyl chain are not critical for binding, the pyrophosphate, MurNac and GlcNac moieties of lipid II are important for recognition by MurJ"

C55-P = UP, since MurJ doesn't bind UP, it is unlikely that it binds UPP.

Even though MurJ binds the phosphate portion of Lipid II, binding to the sugars likely ensures specificity (and contributes to binding energy). We can't assume that all phosphates bind MurJ (then MurJ might also accept Lipid I as a substrate, which would be problematic).

I think lines 469-473 should be removed, they are not essential.

We removed this part, as suggested.

Reviewer #2 (Remarks to the Author):

The authors have fully addressed the comments that I made in my previous report and have made significant changes that have improved the manuscript.

While I do not have any further major concerns, on reading the revised manuscript I have a number of more minor comments that the authors may wish to consider.

- in the introduction, lines 41-43 "most of the predictive models..." - this statement seems to contradict the examples presented above since only 1/3 of these models actually did use the ribosome growth laws as stated here.

We agree that the statement may have been misleading, and now reworded as follows (lines 40-43): "However, to date most of the predictive models for drug-target interactions focussed on translation-inhibiting antibiotics, the description of which is facilitated by a well-established theoretical framework for the partitioning of ribosomes within bacterial cells (4, 5, 6)."

- the 5 antibiotics that are studied are briefly listed in the introduction, but it is only stated that they act against intermediates in the lipid II cycle, whereas in fact it is crucial to know that they act against different intermediates, and which ones. I feel that a little more information here, and also a reference to the diagram of Figure 1a, would be useful.

We thank the reviewer for this comment and included the following descriptions:

Line 78-82: "This is achieved by either targeting the activity of the involved enzymes, e.g. PBPs (inhibited by beta-lactams) and *MraY* (inhibited by tunicamycin), or by directly sequestering the intermediate substrates of the lipid II cycle, e.g. UP (sequestered by frulimicin), UPP (sequestered by bacitracin) or lipid II (sequestered by ramoplanin, vancomycin and nisin), see Fig. 1a for an illustration and (8, 20) for reviews."

Line 91-93: "In particular, we focus on antibiotics targeting different intermediates of the lipid II cycle (substrate-sequestering antibiotics), i.e. bacitracin, frulimicin, ramoplanin, vancomycin and nisin (Fig. 1a), which are active against a broad range of Gram-positive bacteria."

- lines 168-9 "we fixed ... to the experimental values" more accurate here would be to say they were fixed to values obtained from the literature, with references.

Done.

- line 178 "all biochemical...constraints" - this seems exaggerated, I would remove the "all"

Done.

- line 180 "biased" seems a confusing way to put this - "asymmetric" might be clearer.

Done.

- lines 205-6 "concentration per surface area" should be surface density, or surface concentration.

Done.

- line 213 "we find that the" - this is confusing as it sounds as though the authors have measured it, whereas it is actually from the literature. Better would be "literature suggests that the"

Done.

- line 223 "taken this and all existing" is overblown and should be toned down

Toned down: "Taken together, the most parsimonious ..."

- line 244 should contain a reference to the diagram in Fig 1a.

Done.

- line 275 "massive" - this seems exaggerated. Massive compared to what?

We removed the word "massive".

- line 315-6 it would be useful to state in more detail why the authors "wanted to derive an intuitive mathematical formula" - what are the advantages of this?

We now provide an explanation, as follows: "Next, we wanted to derive an intuitive mathematical formula describing how antibiotic susceptibility depends on the pool size of the targeted lipid carrier, thereby rationalizing the origin of the *in vivo* efficacy gap."

- line 318 I appreciate the authors' point that the supplementary text contains a discussion of the level of realism of the analytical model, but I think this should also be explained in the main text, ie rather than just pointing to the supp text one could write "see the supp text for a discussion of the realism of this analytical model"

We now refer to the Methods section, in which the derivation of the simplified model is laid out.

- the buffering effect which is discussed extensively in the latter part of the results is a key result, but it is not necessarily so clear what is meant by this effect from the text. The authors could consider adding a simple diagram to illustrate what is meant by the buffering,

We agree that the description of the buffering effect was not clear enough, but instead of providing another figure (which we feel would be redundant to Fig. 3) we improved our wording, as follows:

“Here $\tilde{K}_G = \frac{[R]}{[T]} \approx \frac{k_{-1}}{k_1}$ describes the ratio between the size of the bactoprenol reservoir (serving as a buffer) and the size of the carrier target in the absence of antibiotic (Fig. 4b). For example, if the buffering reservoir is small compared to the target pool ($\tilde{K}_G \ll 1$), the model predicts only a marginal shift in the IC_{50} ($IC_{50} \approx \tilde{K}_D$), indicating that in this case 50% of the total bactoprenol carriers are easily sequestered by an antibiotic concentration equal to the *in vivo* \tilde{K}_D value. In contrast, if the buffering reservoir is large compared to the target pool ($\tilde{K}_G \gg 1$), an antibiotic concentration equal to the *in vivo* \tilde{K}_D value only sequesters a small amount of the overall bactoprenol carrier level, leading to substantial shifts in IC_{50} ($IC_{50} \gg \tilde{K}_D$).”

- line 57 "is" -> "are"

Done.

- line 82 "if the involves" -> "of the involved"

Done.

- line 250 "two antibiotics" -> "the two antibiotics"

Done.

- line 407 "not always confer" -> "do not always confer"

Done.

Reviewer #3 (Remarks to the Author):

The authors have satisfactorily addressed all of my concerns